**Dissolved Inorganic Nutrients in the Western Mediterranean Sea (2004-2017)**
Malek Belgacem[1,2], Jacopo Chiggiato[1,*], Mireno Borghini[1], Bruno Pavoni[2], Gabriella Cerrati[3],
Francesco Acri[1], Stefano Cozzi[4], Alberto Ribotti[5], Marta Álvarez[6], Siv K. Lauvset[7], Katrin Schroeder[1]
[1] CNR-ISMAR, Arsenale Tesa 104, Castello 2737/F, 30122 Venezia, Italy
[2] Dipartimento di Scienze Ambientali Informatica e Statistica, Università Ca' Foscari Venezia,
Campus Scientifico Mestre, Italy
[3] ENEA, Department of Sustainabiliy, S. Teresa, Marine Environmental center, 19032 Pozzuolo di
Lerici (SP), Italy
[4] CNR-ISMAR, Area Science Park – Basovizza, 34149 Trieste, Italy
[5] CNR-IAS, Loc. Sa Mardini snc, Torregrande, 9170 Oristano, Italy
[6] Instituto Español de Oceanografía, IEO, A Coruña, Spain
[7] NORCE Norwegian Research Centre, Bjerknes Centre for Climate Research, 5007 Bergen, Norway
*Corresponding author's email: jacopo.chiggiato@ismar.cnr.it
**Abstract**
Long-term time-series are a fundamental prerequisite to understand and detect climate shifts and
trends. Understanding the complex interplay of changing ocean variables and the biological
implication for marine ecosystems requires extensive data collection for monitoring, hypothesis testing
and validation of modelling products. In marginal seas, such as the Mediterranean Sea, there are still
monitoring gaps, both in time and in space. To contribute to filling these gaps, an extensive dataset of
dissolved inorganic nutrient observations (nitrate,  phosphate, ⁻ and silicate) has been collected
between 2004 and 2017 in the Western Mediterranean Sea and subjected to rigorous quality control
techniques to provide to the scientific community a publicly available, long-term, quality controlled,
internally consistent biogeochemical data product. The data product includes 870 stations of dissolved
inorganic nutrients, including temperature and salinity, sampled during 24 cruises. Details of the
quality control (primary and secondary quality control) applied are reported. The data are available in
PANGAEA (https://doi.org/10.1594/PANGAEA.904172, Belgacem et al. 2019)
**Keywords:** Mediterranean Sea, Dissolved Inorganic Nutrient, biogeochemistry.
**1    Introduction**
Dissolved inorganic nutrients play a crucial role in marine ecosystem functioning. They serve as
regulators of ocean biological productivity, and are trace elements for biogeochemical cycling as well
as  for natural and anthropogenic sources and transport processes (Bethoux, 1989; Bethoux et al.,
1992). They  are also non-conservative tracers, since their distribution vary according to both
biological (such as primary production and respiration) and physical (such as convection, advection,
mixing and diffusion) processes. Very schematically, inorganic nutrients are continuously consumed
by phytoplankton (due to primary production) in the sea surface and regenerated in the mesopelagic
layer by bacteria and animals (due to respiration). Moreover, the sinking of organic matter and its
decomposition increases the nutrient concentrations in the intermediate and deep-water masses over
time.  To identify the limiting factors for biological production in the oceans, we need to understand
the underlying chemical constraints and especially the macro- and micronutrients spatial and temporal
variations. Dissolved inorganic nutrients may be used as tracers of water masses like salinity and
temperature, to assess mixing processes, and to understand the biogeochemical circumstances of their
formation regions. Understanding the complex interplay of changing ocean variables and the
biological implication for marine ecosystems is a difficult task and requires not only modelling, but
also extensive data collection for monitoring, hypothesis testing and validation. Monitoring gaps still
remain in both in time and space, especially for marginal seas such as the Arctic Ocean or the
Mediterranean Sea.
The Mediterranean Sea has been identified as a region significantly affected by ongoing climatic
changes, like warming and decrease in precipitation (Giorgi, 2006). In addition, it is a region
particularly valuable for climate change research because it behaves like a miniature ocean (Bethoux

et al., 1999) with a well-defined overturning circulation characterized by spatial and temporal scales much shorter than for the global ocean, with a turnover of only several decades. Being an intercontinental sea, and subjected to more terrestrial nutrient inputs (river runoff, submarine groundwater discharge) and atmospheric deposition, the Mediterranean Sea has a nitrate to phosphate N:P ratio that is anomalously high compared to the "classical" world's oceans Redfield ratio, indicating a general P-limitation regime, which becomes stronger along a west-to-east gradient. The Mediterranean Sea is therefore a potential model to study global patterns that will be experienced in the next decades worldwide, not only regarding ocean circulation, but also the marine biota (Lejeusne et al., 2010). Several environmental variables can act as stressors for marine ecosystems, by which climatically driven ecosystem disturbances are generated (Boyd, 2011). These changes affect, among others, the distribution of biogeochemical elements (including inorganic nutrients) and the functioning of the biological pump and $CO_2$ regulation.

Within this context, the aim of this paper is to compile an extensive dataset of dissolved inorganic nutrient observations (nitrate, phosphate, and silicate) collected between 2004 and 2017 in the Western Mediterranean Sea (WMED), to describe the quality control techniques and to provide the scientific community with a publicly available, long-term, quality controlled, and internally consistent biogeochemical data product, contributing to previously published Mediterranean Sea datasets like the MEDAR/Medatlas (time period:1908–1999), (Fichaut et al., 2003) and the Mediterranean Sea – Eutrophication and Ocean Acidification aggregated datasets v2018 (time period: 1911-2017) provided by EMODnet Chemistry (Giorgetti al.,2018) available at https://www.seadatanet.org/Products/Aggregated-datasets.

Both original and quality-controlled data are available in PANGAEA:

https://doi.org/10.1594/PANGAEA.904172Coverage: 44°N-35°S; 6°W-14°E

Location Name: Western Mediterranean Sea

Date start: May 2004

Date end: November 2017

## 2  Dissolved inorganic nutrient data collection

### 2.1. The CNR dissolved inorganic nutrient data in the WMED

Long-term time-series, such as the OceanSites global time series (www.oceansites.org), are a
fundamental prerequisite to understand and detect climate shifts and trends. However, biogeochemical
time-series are still limited to the northern Western Mediterranean Sea (MOOSE network, Coppola et
al., 2019). Yet, inorganic nutrients in the Mediterranean Sea has received more attention in recent
years, and various datasets have been compiled to understand its unique characteristics such as the one
build by the PERSEUS  project Consortium ("Policy-oriented marine environmental research in the
southern European seas" - EU FP7 project GA #287600), that included 100 cruises collected  during
the project's lifetime, in addition to those from other projects like SESAME, EU FP7 project GA
#GOCE-036949), and data products such as the MEDAR/Medatlas. In addition to that, the data
assembly system EMODnet Chemistry, a leading infrastructure supported by pan-European directorate
General MARE set up (Martin Miguez et al., 2019, Tintoré et al.,2019).
The dataset presented here consists of 24 oceanographic cruises (Fig. 1, Table 1a and Table 1b)
conducted in the WMED on board of research vessels run by the Italian National Research Council
(CNR) and the Science and Technology Organisation Centre for Maritime Research and
Experimentation (NATO-STO CMRE). All cruises were merged into a unified dataset with 870
nutrient stations and ~ 9666 data points over a period of 13 years (2004-2017). The overall spatial
distribution of the stations covers the whole WMED, but the actual distribution strongly varies
depending on the specific cruise and most of the data are collected along sections. At all stations,
pressure, salinity and temperature were measured with a CTD-rosette system consisting of a CTD SBE
911 plus and a General Oceanics rosette with 24 12L Niskin Bottles. Temperature measurements were
performed with the SBE-3/F thermometer with a resolution of $10^{-3}$ °C; conductivity measurements
were performed with the SBE-4 sensor with a resolution of $3 \cdot 10^{-4}$ S/m. The probes were calibrated
before and after each cruise. During all CNR cruises, redundant sensors were used for both
temperature and salinity measurements.
Seawater samples for dissolved inorganic nutrient measurements were collected during the CTD up-
cast at standard depths (with slight modifications according to the depth at which the deep chlorophyll
maximum was detected). The standard depths are usually 5, 25, 50, 75, 100, 200, 300, 400, 500, 750,
1000, 1250, 1500, 1750, 2000, 2250, 2500, 2750, 3000 m. No filtration was employed, nutrient
samples were immediately stored at $-20$ °C. Note that sample storage and freezing duration varied
greatly from one cruise to another (Table 3 shows cruises where this exceeded 1 year).

## 2.2. Analytical methods for inorganic nutrients

For all cruises, nutrient determination (nitrate, orthosilicate and orthophosphate) was carried out
following standard colorimetric methods of seawater analysis, defined by Grasshoff et al. (1999) and
Hansen and Koroleff (1999). For inorganic phosphate, the method is based on the reaction of the ions
with an acidified molybdate reagent to yield a phosphomolybdate heteropoly acid, which is then
reduced to a blue-colored compound (absorbance measured at 880 nm). Inorganic nitrate is reduced
(with cadmium granules) to nitrite that react with an aromatic amine leading to the final formation of
the azo dye (measured at 550 nm). Then, the nitrite separately determined must be subtracted from the
total amount measured to get the nitrate concentration only. The determination of dissolved silicon is
based on the formation of a yellow silicomolybdic acid reduced with ascorbic acid to blue-colored
complex (measured at 820 nm).
Nutrient analysis was performed in three laboratories. From 2004 to 2013, all cruises nutrients were
analysed by ENEA, while for those of 2015 (cruise #23) and 2017 (cruise #24), nutrient
concentrations were analysed by CNR-ISMAR. Referring to Table 1S, four different models of
autoanalyzer were used. Measurements from the autoanalyzer were reported in $\mu$mol L$^{-1}$. Inorganic
nutrient concentrations were converted to the standard unit $\mu$mol kg$^{-1}$, using sample salinity from CTD
and a mean laboratory analytical temperature of 20°C. Data from nutrient analysis were then merged
to ancillary CTD bottle data.
**2.3. Reference inorganic nutrient data**
In addition to the data collected during the above-mentioned cruises, and in order to perform the
secondary quality control (described below), we identified five reference cruises (Table 2), based on
their spatial and temporal distribution and the reliability of the measurements (see Fig. 2 –Table.3S
Fig.1S). Cruises 06MT20110405 and 06MT20011018 are the only two Mediterranean cruises included
in the publicly available Global Ocean Data Analysis Project version 2 (GLODAPv2, Olsen et al.
2016). These cruises, conducted on board the R/V Meteor, provide a reliable reference because
nutrient analysis strictly followed the recommendation of the World Ocean circulation experiment
(WOCE) and the GO-SHIP protocols (Hydes et al., 2010; ,Tanhua et al., 2013). Cruises
29AH20140426 and 48UR20070528 are to be included in the CARIMED data product (personal
communication by M. Álvarez, in preparation but not yet available) and have undergone rigorous
quality control following GLODAP routines. Finally, 29AJ20160818 was carried out in the framework
of the MedSHIP programme (Schroeder et al., 2015) and its data are available at
https://doi.org/10.1594/PANGAEA.902293 (Tanhua, 2019).
**3   Quality Assurance and quality control methods**
Combining inorganic nutrient data from different sources, collected by different operators, stored for
different amounts of time, and analysed by multiple laboratories, is not a straightforward task. This is
widely recognized in the biogeochemical oceanographic community. Since the 1990s, several studies
and programmes (e.g. World Ocean Database, World Ocean Atlas, WOCE) have been devoted to
facilitate the exchange of oceanographic data and develop quality control procedures to compile
databases by the estimation of systematic errors (Gouretski and Jancke, 2000) to increase the inter-
comparability, generate consistent data sets and accurately observe the long-term change.
An example of a first quality control procedure is the use of reference materials that are available for
salinity (IAPSO, salinity standard by OSIL) and temperature (SPRT, Standard Platinum Resistance
Thermometer). As for the inorganic carbon, total alkalinity (Dickson et al., 2003) and inorganic
nutrients (Aoyama et al., 2016), certified reference materials (CRM) have been recently made
applicable for oceanographic cruises. However, since CRM are not always available or used for
biogeochemical oceanographic data, Lauvset and Tanhua (2015) developed a secondary quality
control tool to identify biases in deep data. The method suggests adjustments that reduce cruise to
cruise biases, increase accuracy and allow for the inter-comparison between data from various sources.
This approach, based on a crossover and inversion method (Gouretski and Jancke, 2000; Johnson et
al., 2001), was used to generate the CARbon IN Atlantic ocean (CARINA, see Hoppema et al., 2009),
GLODAPv2.2019 (Olsen et al., 2019) and PACIFICA (Suzuki al al.,2013) data products.
**3.1 Primary Quality control**
Each individual cruise was first subjected to a primary quality control (1$^{st}$ QC) that included a check of
apparent and extreme outliers in CTD salinity, nitrate, phosphate and silicate. Each parameter included
a quality control flag, following standard WOCE flags (Table 3). Surface, intermediate and deep layer
were evaluated separately because nutrient observations evolve differently in each layer. The
coefficient of variation (CV, defined as standard deviation over mean) was computed for each depth
layer. Coefficients of variation in the surface (0-250 db) layer were high (nitrate CV=1.16, phosphate
CV=1.005, silicate CV=0.75) due to air-sea interaction (Muniz et al., 2001) occurring in this layer
rendering it difficult to flag. These influences are of reduced importance in the intermediate (250-1000
db) layer (nitrate CV=0.23, phosphate CV=0.31, silicate CV=0.24) and the deep (>1000 db) layer
(nitrate CV=0.15, phosphate CV=0.22, silicate CV=0.14), decreasing the total variance. Flags in the
upper and intermediate layer were thus set based on outliers within pressure ranges defined according
to standard pressures (0-10, 10-30, 30-60, 60-80, 80-160, 160-260, 260-360, 360-460, 460-560, 560-
1000 db).
Below 1000 db, flagging included an inspection of nitrate to phosphate (N:P) and nitrate to silicate (N:
Si) ratios. The Median and Median Absolute Deviation (MAD) was computed by classes of pressure:
we considered as outlier any atypical observation and any value that departs from the median by more
than three MADs in the different pressure ranges for each cruise.
An overview of the nutrient distribution is provided with scatter plots, showing also the flagged
measurements (Fig. 3). Each measurement was flagged 2 ("Acceptable/ measured") or flagged 3
("Questionable"): 4.1% of nitrate data, 3.37% of phosphate data, 3.16% of silicate data, and 0.07% of
CTD salinity data were considered outliers and flagged 3. As highlighted by Tanhua et al. (2010), the
primary QC can be subjective depending on the expertise of the person flagging the data, thus flagging
could bring in some uncertainties.
In order to have a first assessment of the precision of each cruise measurements, the standard deviation
of observations deeper than 1000 db was calculated along with averages and standard deviations for
each cruise and by subregions to have an overview about nutrient content variability in the deep layer
and about the observations spatial spread  of individual cruises (Table 4). Following the subdivision of
Manca et al. (2004), the WMED has been divided into subregions (Fig.2S, Table 2S) according to the
general circulation patterns (details in Manca et al.,2004). Table 4 displays the comparison of standard
deviation of deep measurements for each cruise and within subregions. The overall standard deviation
between cruises in the deep layer varied between 0.51 and 1.41 $\mu$mol kg$^{-1}$ for nitrate, between 0.1 and
1.64 $\mu$mol kg$^{-1}$ for silicate and between 0.025 and 0.078 $\mu$mol kg$^{-1}$ for phosphate. Regional standard
deviation of nitrate measurements below 1000 db varied between 0.08 $\mu$mol kg$^{-1}$ in the Gulf of Lion
(DF2) with cruise #9 and 1.6 $\mu$mol kg$^{-1}$ in the Balearic Sea (DS2) observations of cruise #14.
Phosphate lowest regional standard deviation was 0.01 $\mu$mol kg$^{-1}$ found in the observations of cruise
#9 in Gulf of Lion (DF2), cruise #10 in Balearic Sea (DS2) and Algerian West (DS3), cruise #14 and
cruise # 15 in Tyrrhenian South (DT3), cruise #18 in Algero-Provençal (DF1) and Sardinia Channel
(DI1) while the highest standard deviation was 0.1 $\mu$mol kg$^{-1}$ in the observations of cruise #12 in
Algerian West (DS3). As for silicate, the lowest standard deviation was 0.02 $\mu$mol kg$^{-1}$ observed in
cruise #9 measurements of Gulf of Lion subregion (DF2) and the highest deep standard deviation was
observed in cruise #6 in its all subregions together with cruise #5 measurement in Tyrrhenian North
(DT1) with 1.83 $\mu$mol kg$^{-1}$ standard deviation.
Cruises #3, #6 and #9 had the largest spatial extension (see right side of Fig. 9) with a high number of
samples over more than seven subregions (Table 4), the geographical variability of the distribution in
dissolved inorganic nutrients results thus in the largest standard deviations. Conversely, cruises with
smaller spatial coverages have lower standard deviations. Therefore, a relatively small spatial
coverage and high standard deviation is considered as indicative of data with low precision (Olsen et
al., 2016). This applies to cruises #1, #5, and #16. Despite the small spatial coverage, samples of
nitrate and phosphate of cruise #5 have an overall standard deviation of 1.35 $\mu$mol kg$^{-1}$ and 0.07 $\mu$mol
kg$^{-1}$, respectively, a high standard deviation pointed out also in the regional standard deviation of deep
measurements in Tyrrhenian North (DT1) and South (DT3) . Cruise #1, with few stations in
Tyrrhenian North (DT1) and South (DT3) subregions and 21 samples below 1000 db, has an overall
standard deviation of 1.25 $\mu$mol kg$^{-1}$ for nitrate, 0.06 $\mu$mol kg$^{-1}$ for phosphate and 1.64 $\mu$mol kg$^{-1}$ for
silicate. The regional standard deviation was relatively high for nitrate (0.51-1.32$\mu$mol kg$^{-1}$),
phosphate (0.02-0.065$\mu$mol kg$^{-1}$) and silicate (0.53-1.83$\mu$mol kg$^{-1}$). A comparison with the deviations
from e.g. cruise # 2, carried out in the same year and e.g. cruise #17 (with a similar cruise track),
confirms the lower precision of the data of cruise #1. Similar considerations apply to the quality of
nitrate samples (0.87-1.02 $\mu$mol kg$^{-1}$) and silicate (0.87-0.9 $\mu$mol kg$^{-1}$) from cruise #16, covering a
small area in Tyrrhenian North (DT1) and South (DT3), compared to cruise #17, carried out in the
same regions (right side of Fig. 9 and Table 4).
Deep silicate measurements of cruise #6 have twice the overall standard deviation of silicate data of
cruise #8 from the same year. Adding to that, in the seven subregions, the regional standard deviation
of deep silicate observations was the highest, between 1.04-2 $\mu$mol kg$^{-1}$ which was relatively high
compared to the surrounding cruises that have observations in the same subregions. This is again
suggestive of the limited precision. On the other hand, trying to explain the source of relatively high
standard deviations in specific cruises is not always straightforward, as they could stem from a variety
of sources, sampling, conservation and analysis. The bottom water in the WMED exhibits a high
nutrient content below 1000 db (Table 4), due to the longer residence time. Dividing the WMED into
subregions, has effectively removed the natural spatial change in nutrients, making the interpretation
of the standard deviation a matter of the precision of the measurements only.
In Table 4, deep averages by subregions showed that overall nutrient concentration fluctuated around
7.4 ±0.9$\mu$mol kg$^{-1}$ for nitrate, 0.3 ±0.06 $\mu$mol kg$^{-1}$ for phosphate and 7.7 ±0.8$\mu$mol kg$^{-1}$ for silicate,
similar findings were reported by Manca et al. (2004). Comparing cruise averages in each region
enabled the identification of "suspect" cruises. Cruise #24 has the lowest deep average in nitrate in
Algero-Provençal (DF1), Tyrrhenian North (DT1) subregions and Sardinia Channel (DI1). As for
silicate of cruises #24 and #16 was very low compared to the overall regional average in Liguro-
Provençal (DF3) and Tyrrhenian South (DT3) subregions. Deep average of phosphate did not show
any outlier cruises in all subregions. Different reasons could explain the low precision in the samples,
freezing is one. Although it is a valid preservation method (Dore at al.,1996), the error is higher when
samples were not analysed immediately (Segura-Noguera et al., 2011), so the storage time could
influence.
**3.2 Secondary Quality control: the crossover analysis**
The method used to perform the secondary QC on the WMED dissolved inorganic nutrient dataset
makes use of the quality-controlled reference data, and the crossover analysis toolbox developed by
Tanhua (2010a) and Lauvset and Tanhua (2015). The computational approach is based on comparing
the cruise data set to a high-quality reference data set to quantify biases, described in detail in Tanhua
et al. (2010b). Here, we summarize the technique with emphasis on inorganic nutrients. The first step
consisted of selecting reference data, as described in section 2.3. The second step is the crossover
analysis that was carried out using a MATLAB Toolbox (available online: https://cdiac.ess-
dive.lbl.gov/ftp/oceans/2nd_QC_Tool_V2/) where crossovers are generated as difference between two
cruises using the "running cluster" crossover routine. Each cruise is thus compared to the chosen set of
reference cruises. For each crossover, samples deeper than 1000 db are selected within a predefined
maximum distance set to 2°arc distance, defined as a crossing region, to ensure the quality of the
offset with a minimum number of crossovers and to minimize the effect of the spatial change. The
reason to select measurements deeper than 1000 db, is to remove the high frequency variability
associated to mesoscale features, biological activity and the atmospheric forcing acting in the upper
layers, that might induce changes in biogeochemical properties of water masses. On the other hand,
also the deep Mediterranean cannot be considered truly "unaffected" by changes, as it is intermittently
subjected to ventilation (Schroeder et al., 2016; Testor et al., 2018) and the real variability can be
altered in adjusting data. The computational approach takes this into account, since weights are given
to the less variant profile in the crossing region, according to the "confidence" in the determined offset
of the compared profiles (i.e. the weighted mean offset of a given crossover-pair is weighted to the
depth where the offsets of all compared profiles have the smallest variation which indeed is strongly
interlinked with the degree of variance of each profile) (for further details see Lauvset and Tanhua,

266     2015).

Before identifying crossovers, each profile was interpolated using the piecewise cubic Hermite method
and the distance criteria outlined in Lauvset and Tanhua (2015), their Table 1a, detailed in Key et al.
(2004). The crossover is a comparison between each interpolated profile of the cruise being evaluated
and the interpolated profile of the reference cruise. The result is a weighted offset (defined as
difference cruise/reference) and a standard deviation of the offset. The standard deviation is indicative
of the precision; however, it is important to note that this assumption only works because it is a
comparison to a reference, and the absolute offset is indicative of accuracy.
The third step consists in evaluating and selecting the suggested correction factor that was applied to
the whole water column. The correction factor was calculated from the weighted mean offset of all
crossovers found between the cruise and the reference data set, involving a somewhat subjective
process.
For inorganic nutrients, offsets are multiplicative so that a weighted mean offset > 1 means that the
measurements of the corresponding cruise are higher than the measurements of the reference cruise in
the crossing region and applying the adjustment would decrease the measured values. The magnitude
of an increase or a decrease is the difference of the weighted offset from 1. In general, no adjustment
smaller than 2% (accuracy limit for nutrient measurements) is applied (detailed description is found in
Hoppema et al., 2009; Lauvset and Tanhua, 2015; Olsen et al., 2016; Sabine et al., 2010; Tanhua et al.,
2010b).
The last step is the computation of the weighted mean (WM) to determine the internal consistency and
quantify the overall accuracy of the adjusted product (Hoppema et al., 2009; Sabine et al., 2010;
Tanhua et al., 2009), with the difference that our assessment is based on the offsets with respect to a
set of reference cruises. This WM reflects the absolute weighted mean offset of the data set compared
to the reference data set, hence the smaller the WM the higher the internal consistency. The accuracy
was computed from the individual absolute weighted offsets. The WM, which will be discussed in
section 4.4., was computed using the individual weighted absolute offset (D) of number of crossovers
(L) and the standard deviation ($\sigma$): WM=$\frac{\sum_{i=1}^{L} D(i)/(\sigma(i))^2}{\sum_{i=1}^{L} 1/(\sigma(i))^2}$

## 4 Results of the secondary QC and recommendations

The results of the secondary QC revealed the necessary corrections for nitrate, phosphate and silicate.
Four cruises were not considered in the crossover analysis: cruises #7 and #11 do not have enough
stations > 1000 db (at least 3 to get valid statistics), while cruises #19 and #21 were outside the spatial
coverage of the reference cruises. Cruises that were not used for the crossover analysis are made
available in the original dataset but were not included in the final data product (see Supplementary
material – Part 2 (A2)).
Overall, we found a total number of 73 individual crossovers for nitrate, 72 for phosphate and 54 for
silicate. An example of the running cluster crossover output is shown in Fig.4. Results of the crossover
analysis is an adjustment factor for each cruise and each nutrient, that are shown in Table 5 and Fig. 5-
6-7. The adjustment factor was calculated from the weighted mean of absolute offset summarized in
Table 6 and Fig. 3S-4S-5S. Table 6 details the improvement of the weighted mean of absolute offset
by cruise prior to and after adjustments, the information is also displayed graphically in Fig. 3S-4S-5S.
Cruises are in chronological order in all figures and tables.
**4.1 Nitrate**
The crossover analysis suggests a significant adjustment for nitrate concentrations on 15 cruises,
between 0.94 and 0.98 (for adjustments <1) and between 1.02 and 1.34 (for adjustments >1) (Table 5
and Fig.5). Offsets suggest that the deep measurements of cruises #1, #3, #4, #5, #6, #8, #12, #13, #15,
#16, #23 and #24 need to be adjusted towards higher concentrations, when compared to the respective
reference (Fig.3S).
Nitrate observations of cruises #2, #9 and #10 on the other hand were higher than the reference cruises
and exhibit variation outside the accepted accuracy limit, thus require a downward adjustment.
Finally, five cruises (#14, #17, #18, #20, and #22) were consistent with the reference data and no
adjustment was necessary. Considering the weighted mean of absolute offset after adjustments shown
in Table 6, two cruises (#5 and #24) required large correction factors but remain outside the accuracy
threshold (Fig. 5). These cruises are considered in detail later (section 4.4).
**4.2 Phosphate**
For phosphate the crossover analysis suggests adjustments for 20 cruises, as shown in Fig. 6. Deep
phosphate measurements of 15 cruises (Table 6) appear to be lower than the respective reference
measurements (i.e. phosphate data of these cruises require an upward adjustment), while the data of
five cruises (#2, #3, #4, #6, #24) are higher (i.e. they need a downward adjustment) (Fig.4S). Applying
all the indicated adjustments, the large offsets of cruises #2, #3, #4, #6, #8, #9, #10, #18, #20, #23 and
#24 are reduced and became consistent with the reference. Cruises #1, #5, #12, #13, #14, #15, #16,
#17, and #22 retain an offset even after applying the indicated adjustment. These cruises are
considered in detail later.
According to Olsen et al. (2016), if a temporal trend is detected in the offsets, no adjustments should
be applied. There is indeed a decreasing trend between 2008 and 2017 in the phosphate correction
factor (Fig. 6), and thus an increasing one in the weighted mean offset (Fig.4S), implying a temporal
increase of phosphate. Therefore, phosphate data of the cruises being part of the trend were not
flagged as questionable, except some cruises that are discussed further in section 4.4.
Comparing phosphate before and after adjustment, the corrections did minimise the difference with the
reference, while the actual variation with time was preserved (Fig.6). The temporal trend towards
higher  phosphate concentrations in the Mediterranean Sea is considered to be real, even though
studies concerning the biogeochemical trends in the deep layers of the WMED are scarce (Pasqueron
et al., 2015). However, this variation could be consistent with the findings of Béthoux et al.(1998,
2002) and the modelling studies by Moon et al. (2016) and Powley et al. (2018) who indeed found  an
increasing trend in phosphate concentrations over time, due to the increase in the atmospheric and
terrestrial inputs.
**4.3 Silicate**
The results of the crossover analysis for silicate suggests corrections for all cruises (Fig.7). The
crossovers indicate that deep silicate measurements are lower in the evaluated cruises than in the

corresponding reference cruises (i.e. they need to be adjusted upward) (Fig.5S). This is likely to be a direct result of freezing the samples before analysis, since the reactive silica polymerizes when frozen (Becker et al., 2019). After applying the adjustment (Table 5), as expected, the offsets are reduced (Table 6), but five cruises (#1, #5, #6, #15, and #16) remain outside the accuracy envelope. Due to the large offsets, these cruises will be discussed further in section 4.4.

### 4.4 Discussion and recommendation

Adjustments were evaluated for each cruise separately. As a general rule, no correction was applied when the suggested adjustment is strictly within the 2% limit (indicated with NA in Table 5). The average correction factors were 1.06 for nitrate, 1.14 for phosphate and 1.14 for silicate, respectively. To verify the results, we re-ran the crossover analysis and re-computed offsets and adjustment factors using the adjusted data (as shown in blue in Fig. 3S-4S-5S and Fig. 5-6-7). Most of the new adjustments are within the accuracy envelope and few are outside the limit, except for the cruises belonging to the above mentioned "phosphate-trend" and the other outlying cruises which are detailed hereafter. By the application of adjustments, the deep-water offsets were reduced. This can be seen in the decrease of the weighted mean offset between the data before adjustments (after $1^{st}$ QC, Fig. 3S-4S-5S, in grey) and the adjusted data (after $2^{nd}$ QC, Fig. 3S-4S-5S, in blue).

Referring to the analysis detailed in section 3.2, the internal consistency of the nutrient data set has improved and increased significantly after the adjustment, from 4% for nitrate, 19% for phosphate and 13% for silicate, to a more unified dataset with 3 % for nitrate, 6 % for phosphate and 3% for silicate.

A comparison between the original and the adjusted nutrient observations is shown in Fig. 8A-B-C, indicating an improvement in the accuracy based on the reference data and a relatively reduced range particularly for phosphate (Fig. 8B). Figure 8. D-E scatterplots show that after the quality control, nutrient stoichiometry slopes obtained from regressions, between tracers along the water column demonstrate a strong coupling and provide a nitrate to phosphate ratio of ~22.09 and a nitrate to

silicate ratio of ~0.94. These values are consistent with nutrient ratios range found in the WMED as reported in Lazzari et al. (2016); Pujo-Pay et al., (2011) and Segura-Noguera et al. (2016). The regression model is more accurate after adjustments with an improved $r^2$ for N:P (from 0.81 to 0.90) and for N: Si (from 0.85 to 0.87).

In the following some details on the adjustment of specific cruises are given:

Cruise #2 [48UR20041006] needed an adjustment of 0.98 for nitrate, 0.9 for phosphate and 1.06 for silicate. Most of the crossover profiles occur in the Tyrrhenian Sea (Tyrrhenian North and Tyrrhenian South subregions). After adjustment, the cruise is inside the 2% envelope.

Cruise #3 [48UR20050412] appeared to be outside the 2% envelope before adjustments. Its offsets with five reference cruises, crossing the Tyrrhenian Sea, Sardinia Channel, Gulf of Lion and Algero-Provençal subregions, showed that nitrate and silicate values to be relatively low, and thus an adjustment of 1.08 and 1.15 was applied respectively. On the other hand, phosphate values were relatively high, and a 0.93 adjustment was applied.

Cruise #4 [48UR20050529] correction factor estimate was based on five crossovers that covered five subregions: Tyrrhenian South, Sardinian Channel, Algerian East and West and the Alboran Sea. Table 4 show that there are no large differences between regional averages within the cruise which justify an adjustment of 1.04 for nitrate, 0.85 for phosphate and 1.183 for silicate.

Cruise #8 [48UR20060928] was adjusted by 1.03 for nitrate, 1.14 for phosphate and 1.1 for silicate, because it showed values to be low compared to four references. After adjustment, the data were inside the acceptable range.

Cruise #9 [48UR20071005] values of nitrate were slightly outside the 2% envelope before adjustments, similar to phosphate and silicate that were lower compared to the reference. The

adjustments of 0.97 for nitrate, 1.14 for phosphate and 1.115 for silicate suggested by the mean offset
against the reference cruises were recommended.
Cruise #13 [48UR20090508] has three crossovers in the common crossing zone that included
Tyrrhenian North, Tyrrhenian South and Sardinia Channel subregions. The crossover suggests that this
cruise has too low values and needs an adjustment of 1.05 for nitrate, 1.33 for phosphate and 1.15 for
silicate.
Cruise #14 [48UR20100430] has a mean offset with four reference cruises that suggests an adjustment
factor of 1.34 for phosphate and 1.123 for silicate. Nitrate did fall within the accuracy envelope; no
adjustment was needed.
Cruise #10 [48UR20080318] has only three crossovers in the Algero-Provençal subregion, showing
that nitrate is too high compared to the reference while phosphate and silicate are slightly lower. We
therefore applied the adjustments of Table 5, since the deep averages in each region (Table 4) did not
show large regional difference.
Cruise #17 [48UR20110421] crossover analysis did not suggest any correction for nitrate; however,
with an offset based on two crossovers in the Tyrrhenian North and South subregions, adjustments
were recommended for phosphate (1.25) and silicate (1.12), for being lower than the reference cruises.
Cruise #18 [48UR20111109] is similar to cruise #17, since it was suggested to adjust phosphate by
1.14 and silicate by 1.09, based on four crossovers in the Tyrrhenian North and South, Sardinia
Channel and Algero-Provençal subregions.
Cruise #20 [48UR20120111] has four crossovers over the Tyrrhenian North and South and Algero-
Provençal subregions. Its measurements were slightly lower than the reference cruises suggesting a
correction factor of 1.17 for phosphate and 1.08 for silicate.

Cruise #22 [48UR20131015] has similar correction factors as cruise #20, based on three crossovers in the Sardinia Channel and Tyrrhenian North and South subregion, with measurements being lower than the reference.

Cruise #23 [48QL20150804] showed nutrient values slightly lower than the reference cruises as well, suggesting small correction factors of 1.02 for both nitrate and phosphate and 1.08 for silicate, a correction factors that were based on offsets with five cruises.

Below, we discuss the recommended flags in the final product (Table 3; see supplementary Materials Part-2 (A2)) assigned for some cruises that needed further consideration, since they required larger adjustment factors:

Cruise #1 [48UR20040526]: The adjusted values are still lower than the reference (Fig.5-6-7-Fig.3S-4S-5S) and are still outside the 2% accuracy range. This cruise had stations in the Sicily Strait, Tyrrhenian North and South and Ligurian East subregions (Fig. 9, right side) and only 4 stations were deeper than 1000 db (those within the Tyrrhenian Sea). The low precision of this cruise has already been evidenced during the 1$^{st}$ QC (section 3.1). We recommend flagging this cruise as questionable (flag 3).

Cruise #5 [48UR20051116]: This cruise took place between Sicily Strait and the Tyrrhenian North and South (Fig. 9, right side). Nitrate, phosphate and silicate data were lower than those from other cruises (#3 and #4) run the same year (Fig. 5-6-7-Fig.3S-4S-5S) and are still biased after adjustments. Considering the limited precision and the low number of crossovers, it is recommended to flag the cruise as questionable (flag 3).

Cruise #6 [48UR20060608]: This cruise had an offset with five cruises giving evidence that adjustments of 1.05 for nitrate, 0.86 for phosphate and 1.26 for silicate are needed. The silicate bias was reduced after adjustment but remained large with respect to the accuracy limit (Fig. 7-Fig. 5S). This cruise has a wide geographic coverage, with stations along 9 sections (Fig. 9, right side).

Considering also the high standard deviation (Table 4), which is partially attributed to the spatial
coverage of the cruise, there is still uncertainty about the quality of the samples. It is recommended to
flag silicate data of cruise #6 as questionable (flag 3).
Cruise #12 [48UR20081103]: Phosphate data have low accuracy with respect to the reference cruises
(Fig. 6-Fig. 4S). This cruise has stations along a longitudinal section from Sicily Strait to the Alboran
Sea, which might explain the large standard deviation of deep phosphate samples (Table 4). Cruise
#12 was given a correction of 1.08 for nitrate, 1.12 for silicate and 1.38 for phosphate. The mean
offset from five crossovers computed within the Tyrrhenian South, Sardinia Channel, Algerian East,
Algerian West and Alboran Sea subregions suggests that this cruise has lower nutrient values than the
reference cruise. After adjustment, cruise #12 is within the acceptable range for nitrate and silicate but
not for phosphate as highlighted in section 3.2. In addition, considering the relatively high number of
stations >1000 db and a plausible trend in phosphate, it is recommended to flag the phosphate data as
good/acceptable (flag 2).
Cruise #15 [48UR20100731]: This cruise has 149 station along a similar track as cruise #12 but shows
larger offsets for phosphate and silicate (Fig. 6-7-Fig. 4S-5S), compared to cruise #12. Considering
that deep silicate data was not of low quality (small standard deviation, see Table 4), and that deep
phosphate fall within the "phosphate-trend" discussed above, these data are flagged good/acceptable
(flag 2).
Cruise #16 [48UR20101123]: The cruise shows large offsets for phosphate and silicate (Fig. 6-7- Fig.
4S-5S), similar to cruise #15. Considering that the overall cruise standard deviation of silicate samples
below 1000 db was relatively high (1.02 over 14 samples, see Table 4), and that it has only one
crossover between the Tyrrhenian North and South subregions (Table 6), and that when comparing
deep regional averages, this cruise had the lowest average silicate value, it is recommended to flag
silicate data of cruise #16 as questionable (flag 3). As for phosphate, the cruise is part of the
"phosphate-trend" and is therefore flagged good/acceptable (flag 2).
Cruise #24 [48QL20171023]: This cruise has the largest offset for nitrate even after adjustment. It is
very likely due to a difference between laboratories (calibration standards) concerning nitrate, which
needs to be flagged as questionable (flag 3) in the final product.
There are several sources of bias in the observation. One of the main reasons for an upward/
downward bias would be the difference in the nutrient's chemical analytical method and the lack of
use of CRM in all cruises as also noted in CARINA (Tanhua et al., 2009) or in the most recent global
comparability study by Aoyama (2020).
Cruises discussed in this section were not removed from the final product but are retained along with
their recommended quality flag (Table 3) detailed above and in the supplementary material – Part 2
(A2)). We have done the evaluation of their overall quality but leave it up to the users how to
appropriately use these data.
**4.5 Product assessment: Comparison with MEDATLAS**
Averages water mass biogeochemical properties have been computed from the adjusted product (Table
7), and compared to the MEDAR/Medatlas annual climatological profiles, downloaded from the
Italian NODC website (http://doga.ogs.trieste.it/medar/) given by Manca et al. (2004), in order to
evaluate and  asses the new product. Since nutrient properties exhibit differences with depths, we
compared average nutrient concentrations of the three main water masses in twelve subregions of the
WMED (Table 7, Fig 2S).
The results of Table 7 compares water mass biogeochemical properties with the reference climatology.
The new product agrees well with the Medatlas climatology. However, there are some distinctions.
The surface layer (0-150db) is characterized by a low nutrient content. The surface nitrate varies
between 0.69 and 2.75 $\mu$mol kg$^{-1}$ with a maximum found in the Ligurian East (DF4) and the minimum
in the Alboran Sea (DS1) subregions, similar values were recorded in the climatology (0.61- 3.00
$\mu$mol kg$^{-1}$). The differences in nitrate averages in the surface layer are observed in the Gulf of Lion
(DF2) where the new product is higher than the climatology and slightly lower in the Liguro-
Provençal (DF3).  As for, the surface content in phosphate, it varied between 0.04 and 0.16 $\mu$mol kg$^{-1}$
with a maximum found in the Ligurian East (DF1) and a minimum in the Alboran Sea (DS1), alike the
Medatlas climatology, where phosphate averages fluctuate between 0.05 and 0.19  $\mu$mol kg$^{-1}$ .The new
product is slightly lower compared to the climatology. As to the average surface in silicate, it varies
between 1.36 and 2.91 $\mu$mol kg$^{-1}$ with a minimum found in the Ligurian East (DF4), the maximum in
the Gulf of Lion (DF2)) while in the climatology, it varied between 1.27 and 2.31 $\mu$mol kg$^{-1}$ (the
minimum in the Ligurian East (DF4) and the maximum in the Alboran Sea (DS1)). The new product is
slightly higher in silicate.
Overall, the differences in the surface layer are observed in the Gulf of Lion (DF2), the Liguro-
Provençal (DF3) and the Ligurian East (DF4) regions which could be due to the intense variability of
the vertical mixing occurring in the northern WMED compared to the other subregions.
In the intermediate layer, averages were computed from the depth of the salinity maximum (S$_{max}$)
±100m from a regional average profile, indicative of the Levantine Intermediate Water (LIW) core.
Nitrate average varied between 4.94 and 9.32 $\mu$mol kg$^{-1}$ where the minimum content was recorded in
Sicily strait (DI3) and the maximum in the Algerian West (DS3) while in the Medatlas climatology,
nitrate was between 5.14 and 8.60 $\mu$mol kg$^{-1}$. In average, the lowest content in nitrate was in the
Tyrrhenian North (DT1) and South (DT3), Sardinia Channel (DI1) and Sicily Strait (DI3) while LIW
of the Gulf of Lion (DF2), Liguro-Provençal (DF3), Ligurian East (DF4), Balearic Sea (DS2), Algero-
Provençal (DF1), Alboran Sea (DS1), Algerian West (DS3) and East (DS4) subregions was relatively
rich in nitrate. Compared to the Medatlas product, though the new product was slightly higher mainly
in the Gulf of Lion (DF2), Ligurian East (DF4) and Balearic Sea (DS2). As for phosphate, LIW
averages showed similar behavior as nitrate, the lowest phosphate content (0.21- 0.27 $\mu$mol kg$^{-1}$) was
observed in the Eastern subregions of WMED (DI3,DI1,  DT3 and DT1), when the maximum
concentrations (0.4-0.37 $\mu$mol kg$^{-1}$) were reported in the Western subregions of the WMED (DS1, DS3
and DS4, DS2 and DF2). The large differences between the two products were in the Ligurian East
(DF4) and the Alboran Sea (DS1), subregions of few numbers of observations.
Concerning silicate, the lowest average concentration (5.25 $\mu$mol kg$^{-1}$) was observed in LIW core of
Sicily Strait (DI3,) and the maximum concentrations (8.66 - 8.77 $\mu$mol kg$^{-1}$) were in the Alboran Sea
(DS1) and Gulf of Lion (DF2), similar values were recorded in the Medatlas climatology (4.86-7.95
$\mu$mol kg$^{-1}$). There are some discrepancies, where the new product was higher particularly in the Gulf
of Lion (DF2), Liguro-Provençal (DF3) and Algerian West (DS3) subregions. This difference is
explained by the limited number of observations within depth range in the new product compared to
the observations used in the climatology in these subregions.
Referring to Manca et al.,(2004), the LIW core salinity values are relatively more pronounced in Sicily
Strait (DI3), Sardinia Channel (DI1) and in the Tyrrhenian South (DT3) and North (DT1) subregions,
where nutrients were lower than the Western subregions (DS3, DS4, DS1 , DF1, DS2, DF4, DF3,
DF2). The averages of nutrient within the LIW core ties well with the Medatlas climatology averages
(Table 7), except in subregions with important vertical mixing.
We have verified also average biochemical properties in the deep layer (below 1500db). The new
product is slightly higher in nitrate averages (7.74 -8.37 $\mu$mol kg$^{-1}$) than the Medatlas climatology
(7.12 - 8.06 $\mu$mol kg$^{-1}$) (Table 7). The largest difference was found in Tyrrhenian South (DT3) and
North (DT1) subregions. This difference could be due to the fact that, we are comparing two different
time periods (2004-2017 and 1908-2001). As for the deep layer phosphate, average concentrations
varied between 0.35 and 0.37 $\mu$mol kg$^{-1}$ and were within the climatology limits (0.31 - 0.40 $\mu$mol kg$^{-1}$).
In all subregions, there was not large differences. Overall, phosphate was in accordance with the
Medatlas climatology. Similar to nitrate, deep average silicate in the new product (8.64 -9.21 $\mu$mol kg$^{-1}$
$^{1}$) was higher than the climatology (7.51 to 9.04 $\mu$mol kg$^{-1}$). The largest difference in average silicate
was observed in the Tyrrhenian North (DT1), South (DT3) and Liguro-Provençal (DF3) subregions.
We then used the Root Mean Squared Error (RMSE) as statistical index to quantify the difference
between averaged regional profiles from the new product and Medatlas product. The climatology
annual profiles were interpolated to the regional average profiles of the new product, and the average
RMSE for each layer and subregion was calculated.   Fig. 10 shows the regional evolution of RMSE in
the main water masses for the three nutrients. For nitrate (Fig. 10 A), the RMSE in the surface layer
varied between 0.12 $\mu$mol kg$^{-1}$ (in the Tyrrhenian North (DT1)) and 1.36 $\mu$mol kg$^{-1}$ (in the Gulf of
Lion (DF2)); in the intermediate layer, the RMSE was between 0.07 $\mu$mol kg$^{-1}$ (in the Sardinia
Channel (DI1)) and 2.35 $\mu$mol kg$^{-1}$ (in the Gulf of Lion (DF2)), and was lower in the deep layer,
between 0.11 $\mu$mol kg$^{-1}$ (in the Algerian East (DS4)) and 0.79 $\mu$mol kg$^{-1}$ (the Gulf of Lion (DF2)). The
RMSE decreases in the Algerian East (DS4), Tyrrhenian North (DT1), Tyrrhenian South (DT3),
Sardinia Channel (DI1) and Sicily Strait (DI3). This illustrates the low difference between the two
products.
For phosphate (Fig. 10 B), the RMSE ranges between 0.0022 $\mu$mol kg$^{-1}$ (in the Tyrrhenian South
(DT3)) and 0.12 $\mu$mol kg$^{-1}$ (in the Ligurian East (DF4)) in the surface layer; and is between 0.003
$\mu$mol kg$^{-1}$ (in the Liguro-Provençal subregion (DF3)) and 0.048 $\mu$mol kg$^{-1}$ (in  the Alboran Sea (DS1))
at intermediate depths, while in the deep layer RMSE varied between 0.0087 (in the Gulf of Lion
(DF2)) and 0.057 $\mu$mol kg$^{-1}$ (in the Tyrrhenian North (DT1)).
Regarding silicate RMSE (Fig. 10 C) in surface, it varied between 0.13 $\mu$mol kg$^{-1}$ (in the Algero-
Provençal subregion (DF1)) and 3.5 $\mu$mol kg$^{-1}$ (in the Ligurian East subregion (DF4)), A lower RMSE
between 0.10 $\mu$mol kg$^{-1}$ (in the Sardinia Channel (DI1)) and 2.54 $\mu$mol kg$^{-1}$ (in the Gulf of Lion (DF2))
was reported in the intermediate layer; the results in deep layer, were  between  0.33 $\mu$mol kg$^{-1}$ (in the
Algerian East (DS4)) and 1.43 $\mu$mol kg$^{-1}$ (in the Liguro-Provençal subregion (DF3)).
The best agreement between the two products was observed in the intermediate and deep layer. The
lowest RMSE was confined to the deep layer in most of the subregions while the highest difference
was found in the surface layer since it is subjected to intense vertical mixing mainly in the northern
WMED. Comparing averages in subregions, showed similar differences in nutrient between the two
products particularly in the Gulf of Lion (DF2), the Liguro-Provençal (DF3), Ligurian East (DF4) and
Algerian East (DS4), due to the relative high variability in nutrient concentrations in these subregions.
These differences are not significant as there is discrepancy on the number of observations used in the
two products. Overall, inorganic nutrients of the new product agree very well with the
MEDAR/Medatlas climatology. The main features of the spatial distribution in the inorganic nutrients
were in accordance with the findings of Manca et al., (2004), where the relative high content in
nutrient was found in the intermediate layer of the Algerian subregions (DF1, DS3, DS4) than in other
subregions ( Table 7). Besides, the highest concentrations in deep layer silicate were reported in the
Algerian subregions in the two products (9.21 $\mu$mol kg$^{-1}$ (DS3) in the new product; 9.04 $\mu$mol kg$^{-1}$
(DS4) in the climatology), which is indicative of the poor regional ventilation and of the longer
residence time of deep water especially in these subregions.

## 5 Final remarks

An internally consistent data set of dissolved inorganic nutrients has been generated for the WMED
(2004-2017). The accuracy envelope for nitrate and silicate was set to 2%, a predefined limit used in
GLODAP and CARINA data products. Regarding phosphate data, these were almost entirely outside
this limit, because of its natural variations and the overall very low concentrations in the WMED, a
highly P-limited basin. Using a crossover analysis (2$^{nd}$ QC toolbox) to compare cruises with respect to
reliable reference data, improved the accuracy of the measurements by bias-minimizing the individual
cruises. The new product was broadly in consistent with the earlier climatology MEDAR/Medatlas.
The publication of a quality-controlled extensive (spatially and temporally) database of inorganic
nutrients in the WMED was timely and fills a gap in information that prevented baseline assessments
on spatial and temporal variability of biogeochemical tracers in the Mediterranean. In combination
with older databases in the same region (e.g. bottle data available in the MEDAR/Medatlas database),
this new data producte will thus constitute a pillar on which the Mediterranean marine scientific

community will be able to build on original research topics on biogeochemical fluxes and cycles and their relation to hydrological changes that occurred in the period covered by the dataset. The dataset is also relevant for the modelling community as it can be used as an independent data product to assess reanalysis products or it can be assimilated in new reanalysis products.

## 6  Data availability

The final product is available as a .csv merged file from PANGAEA, and can be accessed at https://doi.org/10.1594/PANGAEA.904172 (Belgacem et al. 2019).

Ancillary information is in the supplementary materials with the list of variables included in the original and final product. Table 1a and Table 1b summarizes all cruises included in the dataset. The dataset include frequently measured stations and key transects of the WMED with in situ physical and chemical oceanographic observations. As mentioned, two files are accessible, both include oceanographic variables observed at the standard depths (see supplementary Materials Part-2).

- *Original dataset: CNR_DIN_WMED_20042017_original.csv:* This is the original dataset with flag variable for each of the following parameter: CTD salinity, nitrate, phosphate and silicate from the primary quality control (detailed in section 3.1).
- *Adjusted dataset: CNR_DIN_WMED_20042017_adjusted.csv*: This is the product after primary quality control and after applying the adjustment factors from the secondary quality control. Recommendations of section 4.4 are included, as well as quality flags.

**Author contribution:** MB, MA, SL, JC and KS substantially contributed to write the manuscript. SC, GC and FA run the chemical analysis and contributed to the manuscript. MB coordinated the technical aspects of most of the cruises. SC, GC, FA, AR, BP contributed in specific part of the manuscript.

**Acknowledgements.** The data have been collected in the framework of several of national and European projects, e.g.: KM3NeT, EU GA #011937; SESAME, EU GA #GOCE-036949; PERSEUS,

EU GA #287600; OCEAN-CERTAIN, EU GA #603773; COMMON SENSE, EU GA #228344;
EUROFLEETS, EU GA #228344; EUROFLEETS2, EU GA # 312762; JERICO, EU GA #262584;
the Italian PRIN 2007 program "Tyrrhenian Seamounts ecosystems", and the Italian RITMARE
Flagship Project, both funded by the Italian Ministry of University and Research. We thank Sarah
Jutterström from the Swedish Environmental Research institute for the invaluable help in Quality
Control discussions. We would like to express our appreciation to the INOCEN laboratory team at
IEO for their help and collaboration during MB's stay there. The authors are deeply indebted to all
investigators and analysts who contributed to data collection at sea during so many years, as well as to
the PIs of the cruises (S. Aliani, M. Astraldi, M. Azzaro, M. Dibitetto, G. P. Gasparini, A. Griffa, J.
Haun, L. Jullion, G. La Spada, E. Manini, A. Perilli, C. Santinelli, S. Sparnocchia), the captains and
the crews for allowing the collection of this enormous dataset; without them, this work would not have
been possible.

















**References** Aoyama, M., Woodward, E., Malcolm, S., Bakker, K., Becker, S., Björkman, K., Daniel, A., Mahaffey, C., Murata, A., Naik, H., Tanhua, T., Rho, T., Roman, R. and Sloyan, B.: Comparability of oceanic nutrient data, Poster Cluster Community Whitepaper, CLIVAR Open Science Conference on "Charting the course for climate and ocean research", 18-25 September 2016, Qingdao (China), 12 pp., http://hdl.handle.net/10261/17137, 2016.

Aoyama, M.: Global certified-reference-material-or reference-material-scaled nutrient gridded dataset GND13. Earth System Science Data, 12, 487-499, https://doi.org/10.5194/essd-12-487-2020, 2020.

Becker, S., Aoyama, M., Woodward, E.M.S., Bakker, K., Coverly. S., Mahaffey, C., and Tanhua, T.: GO-SHIP Repeat Hydrography Nutrient Manual: The precise and accurate determination of dissolved inorganic nutrients in seawater, using Continuous Flow Analysis methods, In: The GO-SHIP Repeat Hydrography Manual: A Collection of Expert Reports and Guidelines, 56 , http://dx.doi.org/10.25607/OBP-555, 2019.

Belgacem, M., Chiggiato, J., Borghini, M., Pavoni, B., Cerrati, G., Acri, F; Cozzi, S., Ribotti, A., Álvarez, M., Lauvset, S. K., Schroeder, K.: Quality controlled dataset of dissolved inorganic nutrients in the western Mediterranean Sea (2004-2017) from R/V oceanographic cruises. PANGAEA, https://doi.org/10.1594/PANGAEA.904172, 2019.

Bethoux, J. P.: Oxygen consumption, new production, vertical advection and environmental evolution in the Mediterranean Sea, Deep Sea Research, Part A, Oceanographic Research Papers, 36(5), 769–781, doi:10.1016/0198-0149(89)90150-7, 1989.

Bethoux, J. P., Morin, P., Madec, C. and Gentili, B.: Phosphorus and nitrogen behaviour in the Mediterranean Sea, Deep Sea Research, Part A, Oceanographic Research Paper, 39(9), 1641–1654, doi:10.1016/0198-0149(92)90053-V, 1992.

Bethoux, J. P., Gentili, B., Morin, P., Nicolas, E., Pierre, C. and Ruiz-Pino, D.: The Mediterranean

Sea : a miniature ocean for climatic and environmental studies and a key for the climatic funcioning of
the North Atlantic, Progress in Oceanography, 44, 131–146, 1999.
Béthoux, J. P., Morin, P., Chaumery, C., Connan, O., Gentili, B. and Ruiz-Pino, D.: Nutrients in the
Mediterranean Sea, mass balance and statistical analysis of concentrations with respect to
environmental change, Marine Chemestry , 63(1–2), 155–169, doi:10.1016/S0304-4203(98)00059-0,
666 1998.

Béthoux, J. P., Morin, P. and Ruiz-Pino, D. P.: Temporal trends in nutrient ratios: Chemical evidence
of Mediterranean ecosystem changes driven by human activity, Deep  Sea Research Part II Topical
Studies in Oceanography, 49(11), 2007–2016, doi:10.1016/S0967-0645(02)00024-3, 2002.
Boyd, P. W.: Beyond ocean acidification, Nature Geoscience, 4(5), 273–274, doi:10.1038/ngeo1150,
671 2011.

Coppola, L., Raimbault, P., Mortier, L., and Testor, P.: Monitoring the environment in the
northwestern Mediterranean Sea, Eos, 100, https://doi.org/10.1029/2019EO125951, 2019.
Dickson, A. G., Afghan, J. D. and Anderson, G. C.: Reference materials for oceanic CO2 analysis: A
method for the certification of total alkalinity, Marine Chemistry, 80(2–3), 185–197,
doi:10.1016/S0304-4203(02)00133-0, 2003.
Dore, J. E., Houlihan, T., Hebel, D. V., Tien, G., Tupas, L., Karl, D. M.: Freezing as a method of
sample preservation for the analysis of dissolved inorganic nutrients in seawater, Marine
Chemistry, 53(3-4), 173-185, 1996.
Fichaut, M., Garcia, M. J., Giorgetti, A., Iona, A., Kuznetsov, A., Rixen, M. and Group, M.:
MEDAR/MEDATLAS 2002: A Mediterranean and Black Sea database for operational oceanography,
Elsevier Oceanography Series, 69, 645–648, doi:10.1016/S0422-9894(03)80107-1, 2003.
Giorgetti, A., Partescano, E., Barth, A., Buga, L., Gatti, J., Giorgi, G., Iona A., Lipizer, M.,
Holdsworth, N., Larsen, M.M., Schaap, D., Vinci, M., Wenzer, M. :EMODnet Chemistry Spatial Data
Infrastructure for marine observations and related information. Ocean & Coastal Management, 166, 9-
17, 2018.Giorgi, F.: Climate change hot-spots, Geophysical Research Letters, 33(8), 1–4,
doi:10.1029/2006GL025734, 2006.
Gouretski, V. V. and Jancke, K.: Systematic errors as the cause for an apparent deep water property
variability: Global analysis of the WOCE and historical hydrographic data, Progress in Oceanography,
48(4), 337–402, doi:10.1016/S0079-6611(00)00049-5, 2000.
Grasshoff, K., Kremling K., Ehrhardt M.: Methods of seawater analysis (3rd ed.), Weinheim
Press, WILEY-VCH, 203-273, 1999.
Hansen, H. P. and Koroleff, F.: Determination of nutrients, Methods of Seawater Analysis, 159–228,
694 1999.

Hoppema, M., Velo, A., van Heuven, S., Tanhua, T., Key, R. M., Lin, X., Bakker, D. C. E., Perez, F.
F., Ríos, A. F., Lo Monaco, C., Sabine, C. L., Álvarez, M. and Bellerby, R. G. J.: Consistency of
cruise data of the CARINA database in the Atlantic sector of the Southern Ocean, Earth System
Science Data, 1(1), 63–75, doi:10.5194/essd-1-63-2009, 2009.
Hydes, D. J., Aoyama, M., Aminot, A., Bakker, K., Becker,S., Coverly, S., Daniel, A., Dickson, A. G.,
Grosso, O., Kerouel, R., van Ooijen, J., Sato, K., Tanhua, T., Woodward, E. M. S. and Zhang, J. Z.
:Determination of Dissolved Nutrients (N, P, SI) in Seawater With High Precision and Inter-
Comparability Using Gas-Segmented Continuous Flow Analysers. In: The GO-SHIP Repeat
Hydrography Manual: A Collection of Expert Reports and Guidelines. Version 1. (eds Hood, E.M.,
C.L. Sabine, and B.M. Sloyan). IOCCP Report Number 14, ICPO Publication Series Number 134. 87
pp., http://dx.doi.org/10.25607/OBP-555, 2010.
Johnson, G. C., Robbins, P. E. and Hufford, G. E.: Systematic adjustments of hydrographic sections
for internal consistency, Journal of Atmospheric Oceanic Technology, 18(7), 1234–1244,
doi:10.1175/1520-0426(2001)018<1234:SAOHSF>2.0.CO;2, 2001.
Key, R. M., Kozyr, A., Sabine, C. L., Lee, K., Wanninkhof, R., Bullister, J. L., Feely, R. A., Millero,
F. J., Mordy, C. and Peng, T. H.: A global ocean carbon climatology: Results from Global Data
Analysis Project (GLODAP), Global Biogeochem. Cycles, 18(4), 1–23, doi:10.1029/2004GB002247,
713    2004.

Lauvset, S. K. and Tanhua, T.: A toolbox for secondary quality control on ocean chemistry and
hydrographic    data,    Limnology    and    Oceanography    Methods,    13(11),    601–608,
doi:10.1002/lom3.10050, 2015.
Lazzari, P., Solidoro, C., Salon, S. and Bolzon, G.: Spatial variability of phosphate and nitrate in the
Mediterranean  Sea :  A  modeling  approach,  Deep  Sea  Research  Part  I,  108,  39–52,
doi:10.1016/j.dsr.2015.12.006, 2016.
Lejeusne, C., Chevaldonné, P., Pergent-Martini, C., Boudouresque, C. F. and Pérez, T.: Climate
change effects on a miniature ocean: the highly diverse, highly impacted Mediterranean Sea, Trends in
Ecology and Evolution, 25(4), 250–260, doi:10.1016/j.tree.2009.10.009, 2010.
Manca, B., Burca, M., Giorgetti, A., Coatanoan, C., Garcia, M. J., and Iona, A. : Physical and
biochemical averaged vertical profiles in the Mediterranean regions: an important tool to trace the
climatology of water masses and to validate incoming data from operational oceanography. Journal of
Marine Systems, 48(1-4), 83-116, 2004.
Martín Míguez, B., Novellino, A., Vinci, M., Claus, S., Calewaert, J. B., Vallius, H.,Schmitt, T.,
Pititto, P., Giorgetti, A., Askew,N., Iona, S., Schaap, D., Pinardi, N., Harpham,.Q, Kater, B.J.,
Populus, J.,She, J., Vasilev Palazov, A., McMeel, O., Oset, P., Lear,D., Manzella, G.M.R., Gorringe,
P., Simoncelli, S.,Larkin, K., Holdsworth, N., Dimitrios_Arvanitidis  C., Molina-Jack M.E., Chaves-
Montero M.D.M. , Herman, P.M.J., and Hernandez F.: The European marine observation and data
network (EMODnet): visions and roles of the gateway to marine data in Europe. Frontiers in Marine
Science, 6, 2019.
Moon, J., Lee, K., Tanhua, T., Kress, N. and Kim, I.: Temporal nutrient dynamics in the
Mediterranean    Sea    in    response    to    anthropogenic    inputs,    ,    5243–5251,
doi:10.1002/2016GL068788.Received, 2016.
Muniz, K., Cruzado, A., Ruiz De Villa, C. and Villa, C. R. De: Statistical analysis of nutrient data
quality ( nitrate and phosphate ), applied to useful predictor models in the northwestern Mediterranean
Sea, Methodology, 17, 221–231, 2001.
Olsen, A., Key, R. M., Heuven, S. Van, Lauvset, S. K., Velo, A., Lin, X., Schirnick, C., Kozyr, A.,
Tanhua, T., Hoppema, M. and Jutterström, S.: The Global Ocean Data Analysis Project version 2 (
GLODAPv2 ) – an internally consistent data product for the world ocean, , 297–323,
doi:10.5194/essd-8-297-2016, 2016.
Olsen, A., Lange, N., Key,R., Tanhua, T., Alvarez, M. et al.: GLODAPv2.2019 -an update of
GLODAPv2. Earth Syst. Sci. Data, 11 (3), pp.1437 - 1461. ff10.5194/essd-11-1437-2019ff. ffhal-
746   02315662, 2019.

Pasqueron, O., Fommervault, D., Migon, C., Ortenzio, F. D., Ribera, M. and Coppola, L.: Temporal
variability of nutrient concentrations in the northwestern Mediterranean sea ( DYFAMED time-series
station ), Deep. Res. Part I, 100, 1–12, doi:10.1016/j.dsr.2015.02.006, 2015.
Powley, H. R., Krom, M. D. and Van Cappellen, P.: Phosphorus and nitrogen trajectories in the
Mediterranean Sea (1950–2030): Diagnosing basin-wide anthropogenic nutrient enrichment, Progress
in Oceanography, 162, 257–270, doi:10.1016/j.pocean.2018.03.003, 2018.
Pujo-Pay, M., Conan, P., Oriol, L., Cornet-Barthaux, V., Falco, C., Ghiglione, J. F., Goyet, C.,
Moutin, T. and Prieur, L.: Integrated survey of elemental stoichiometry (C, N, P) from the western to
eastern Mediterranean Sea, Biogeosciences, 8(4), 883–899, doi:10.5194/bg-8-883-2011, 2011.
Sabine, C. L., Hoppema, M., Key, R. M., Tilbrook, B., Van Heuven, S., Lo Monaco, C., Metzl, N.,
Ishii, M., Murata, A. and Musielewicz, S.: Assessing the internal consistency of the CARINA data
base in the Pacific sector of the Southern Ocean, Earth System Science Data Discussions, 2(2), 195–
204, doi:10.5194/essd-2-195-2010, 2010.
Schroeder, K., Tanhua, T., Bryden, H., Alvarez, M., Chiggiato, J. and Aracri, S.: Mediterranean Sea
Ship-based Hydrographic Investigations Program (Med-SHIP), Oceanography, 28(3), 12–15,
doi:10.5670/oceanog.2015.71, 2015.
Schroeder, K., Chiggiato, J., Bryden, H. L., Borghini, M. and Ben Ismail, S.: Abrupt climate shift in
the Western Mediterranean Sea, Scientific Reports, 1–7, doi:10.1038/srep23009, 2016.Segura-
Noguera, M., Cruzado, A. and Blasco, D.: The biogeochemistry of nutrients, dissolved oxygen and
chlorophyll a in the Catalan Sea (NW Mediterranean Sea), Sci. Mar., 80(S1), 39–56,
doi:10.3989/scimar.04309.20a, 2016.
Segura-Noguera, M., Cruzado, A., & Blasco, D.: Nutrient preservation, analysis precision and quality
control of an oceanographic database of inorganic nutrients, dissolved oxygen and chlorophyll a from
the NW Mediterranean Sea. Scientia Marina, 75(2), 321-339, 2011.
Suzuki, T., Ishii, M., Aoyama, A., Christian, J. R., Enyo, K., Kawano, T., Key, R. M., Kosugi, N.,
Kozyr, A., Miller, L. A., Murata, A., Nakano, T., Ono, T., Saino, T., Sasaki, K., Sasano, D., Takatani,
Y., Wakita, M., and Sabine, C. L.: PACIFICA Data Synthesis Project, ORNL/CDIAC-159, NDP-092,
Carbon Dioxide Information Analysis Center, Oak Ridge National Laboratory, U. S. Department of
Energy, Oak Ridge, Tennessee, 2013.
Tanhua, T.: Hydrochemistry of water samples during MedSHIP cruise Talpro. PANGAEA,
https://doi.org/10.1594/PANGAEA.902293, 2019.
Tanhua, T.: Matlab Toolbox to Perform Secondary Quality Control (2nd QC) on Hydrographic Data,
ORNL CDIAC-158. Carbon Dioxide Inf. Anal. Center, Oak Ridge Natl. Lab. U.S. Dep. Energy, Oak
Ridge, Tennessee, 158, doi:10.3334/CDIAC/otg.CDIAC_158, 2010a.
Tanhua, T., Brown, P. J. and Key, R. M.: CARINA : Nutrient data in the Atlantic Ocean, Earth
Science Data, 1, 7–24, doi:10.3334/CDIAC/otg.CARINA.ATL.V1.0, 2009.
Tanhua, T., Heuven, S. van, Key, R. M., Velo, A., Olsen, A. and Schirnick, C.: Quality control
procedures and methods of the CARINA database, Earth System Scence Data, 2, 35–49, 2010b.
Tanhua, T., Hainbucher, D., Schroeder, K., Cardin, V., Álvarez, M. and Civitarese, G.: The
Mediterranean Sea system: A review and an introduction to the special issue, Ocean Science, 9(5),
789–803, doi:10.5194/os-9-789-2013, 2013.
Testor, P., Bosse, A., Houpert, L., Margirier, F., Mortier, L., Legoff, H., Dausse, D., Labaste, M.,
Karstensen, J., Hayes, D., Olita, A., Ribotti, A., Schroeder, K., Chiggiato, J., Onken, R., Heslop, E.,
Mourre, B., D'ortenzio, F., Mayot, N., Lavigne, H., de Fommervault, O., Coppola, L., Prieur, L.,
Taillandier, V., Durrieu de Madron, X., Bourrin, F., Many, G., Damien, P., Estournel, C., Marsaleix,
P., Taupier-Letage, I., Raimbault, P., Waldman, R., Bouin, M. N., Giordani, H., Caniaux, G., Somot,
S., Ducrocq, V. and Conan, P.: Multiscale Observations of Deep Convection in the Northwestern
Mediterranean Sea During Winter 2012–2013 Using Multiple Platforms, Journal of Geophysical
Research: Oceans, 123(3), 1745–1776, doi:10.1002/2016JC012671, 2018.
Tintoré, J., Pinardi, N., Alvarez Fanjul, E., Balbin, R., Bozzano, R., Ferrarin, C.,... and Clementi, E.:
Challenges for Sustained Observing and Forecasting Systems in the Mediterranean Sea. Frontiers in
Marine Science, 6, 568, 2019.

 **Figure Captions**

**Figure 1.** Map of the Western Mediterranean Sea showing the biogeochemical stations (in blue) and
the five reference cruise stations (in red).
**Figure 2.** Overview of the reference cruise spatial coverage and vertical distributions of the inorganic
nutrients. Top left: geographical distribution map, top right: vertical profiles of nitrate in $\mu$mol kg$^{-1}$,
bottom left: vertical profiles of phosphate in $\mu$mol kg$^{-1}$, bottom right: vertical profiles of silicate in
$\mu$mol kg$^{-1}$.
**Figure 3.** Scatter plots of (A.) phosphate vs nitrate (in $\mu$mol kg$^{-1}$) and (B.) silicate vs. nitrate (in $\mu$mol
kg$^{-1}$). Data that have been flagged as "questionable" (flag=3) are in red, the colour bar indicates the
pressure (in dbar). The black lines represent the best linear fit between the two parameters, and the
corresponding equations and r$^2$ values are shown on each plot. Average resulting N:P ratio is 20.87,
average resulting N:Si ratio is 1.05 (whole depth).
**Figure 4.** An example of the calculated offset for silicate between cruise 48UR20131015 and cruise
29AJ2016818 (reference cruise). Above: location of the stations being part of the crossover and
statistics. Bottom left: vertical profiles of silicate data in ($\mu$mol kg$^{-1}$) of the two cruises that fall within
the minimum distance criteria (the crossing region), below 1000 dbar. Bottom right: vertical plot of
the difference between both cruises (dotted black line) with standard deviations (dashed black lines)
and the weighted average of the offset (solid red line) with the weighted standard deviations (dotted
red line).
**Figure 5**. Results of the crossover analysis for nitrate, before (grey) and after adjustment (blue). Error
bars indicate the standard deviation of the absolute weighted offset. The dashed lines indicate the
accuracy limit 2% for an adjustment to be recommended.
**Figure 6**. The same as Fig. 5 but for phosphate.

**Figure 7**. The same as Fig. 5 but for silicate.

**Figure 8.** Dataset comparison before (black) and after (blue) adjustment, showing vertical profiles of (A.) nitrate (in $\mu$mol kg$^{-1}$), (B.) phosphate (in $\mu$mol kg$^{-1}$) and (C.) silicate (in $\mu$mol kg$^{-1}$). Scatter plots of the adjusted data from all depths after 1$^{st}$ and 2$^{nd}$ quality control for (D.) phosphate vs nitrate (in $\mu$mol kg$^{-1}$) and (E.) silicate vs. nitrate (in $\mu$mol kg$^{-1}$). The black lines represent the best linear fit between the two parameters, and the corresponding equations and r$^{2}$ values are shown on each plot. Average resulting N:P ratio is 22.09, average resulting N:Si ratio is 0.94 (whole depth).

**Figure 9.** Vertical profiles of the inorganic nutrients in the dataset after adjustments and spatial coverage of each cruise (reference to cruise ID is above each map). The whole WMED adjusted product is shown in black while the data of each individual cruise are shown in blue (flag=2) and green (flag=3).

**Figure 10.** RMSE regional averages of water mass properties computed between the new adjusted product and MEDAR/Medatlas climatology for nitrate (A.), phosphate (B.) and silicate (C.).

**Table captions**

**Table 1a.** Cruise summary table and parameters listed with number of stations and samples. Cruises were identified with an ID number and expedition code ('EXPOCODE' of format AABBYYYYMMDD with AA: country code, BB: ship code, YYYY: year, MM: month, DD: day indicative of cruise starting day).

**Table 1b.** Data sources and links to the reports (accessed June 2020).

**Table 2**. Cruise summary table of the reference cruises collection used in the secondary quality control, collected from 2001 to 2016.

**Table 3**. WOCE flags used in the original data product and in the adjusted product.

**Table 4.** Average and Standard deviations of nitrate, phosphate and silicate measurements by cruise and for each region with number of samples deeper than 1000db included in the $2^{nd}$ QC. Average storage time: the minimum storage time defined as time difference between the cruise ending day and the $1^{st}$ day of the laboratory analysis.

**Table 5.** Summary of the suggested adjustment for nitrate, phosphate and silicate resulting from the crossover analysis. Adjustments for inorganic nutrient are multiplicative. NA: denotes not adjusted, i.e. data of cruises that could not be used in the crossover analysis, because of the lack of stations or data are outside the spatial coverage of reference cruises.

**Table 6.** Secondary QC toolbox results: improvements of the weighted mean of absolute offset per cruise of unadjusted and adjusted data; (n) is the number of crossovers per cruise. The numbers in red (less than 1) indicate that the cruise data are lower than the reference cruises. NA: not adjusted.

**Table 7.** Water mass properties and regional average concentrations of inorganic nutrients: comparison between the new adjusted product and the MEDAR/Medatlas climatology (with standard deviations and number of observations in brackets).

**Figure 1**

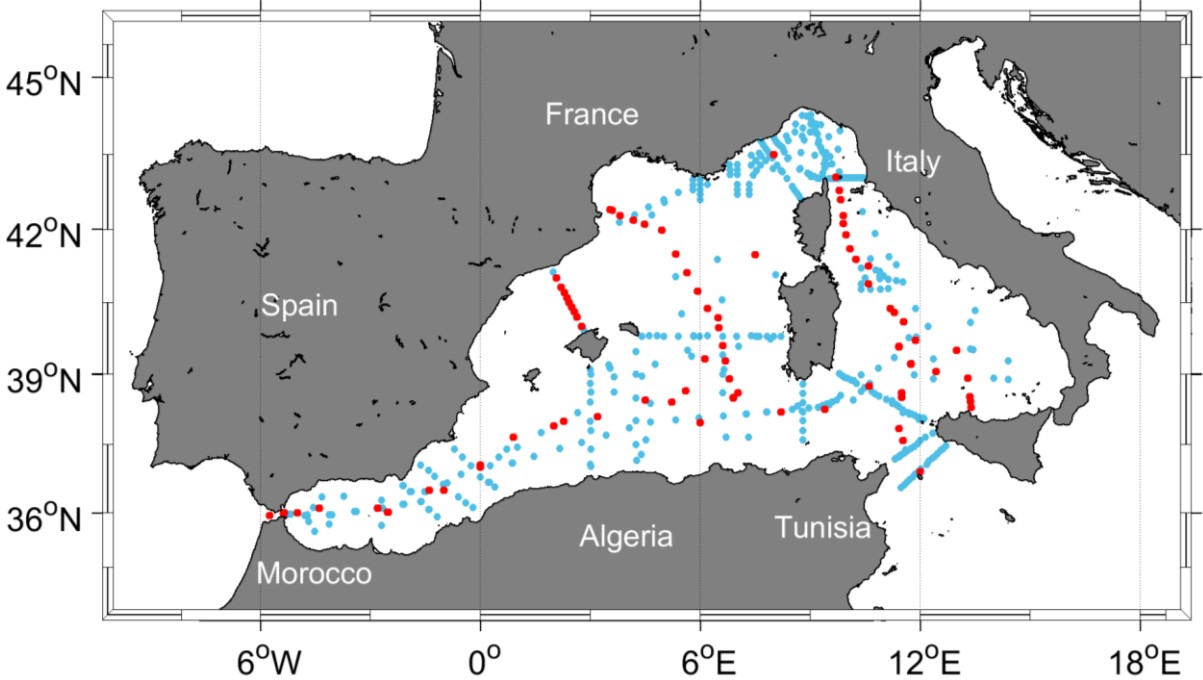












**Figure 2**

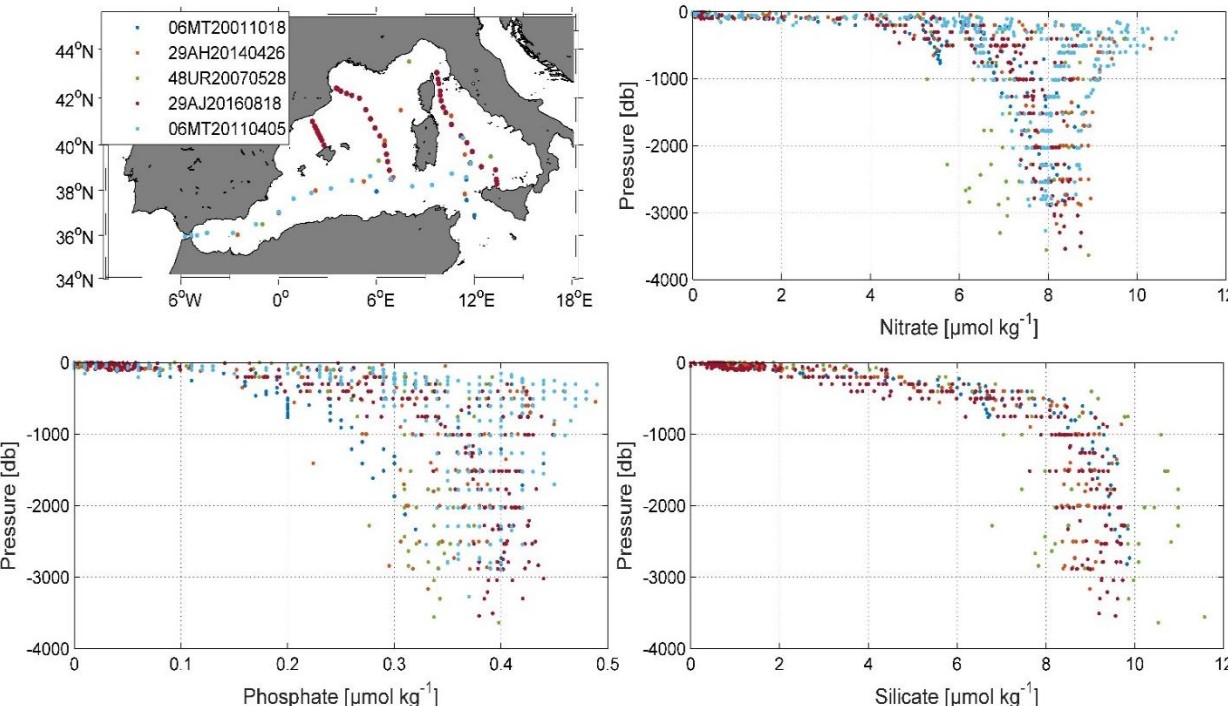















**Figure 3**

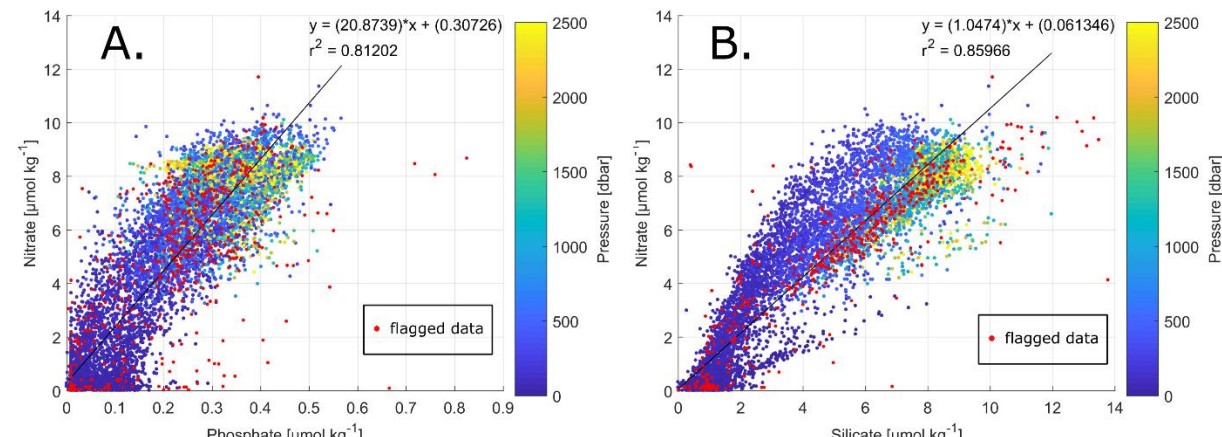







**Figure 4**




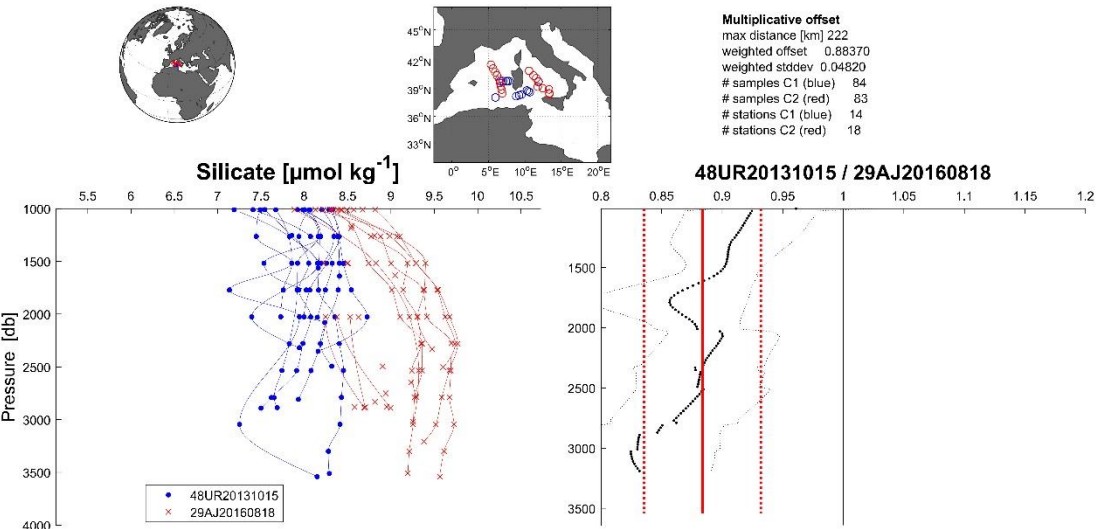


**Figure 5**

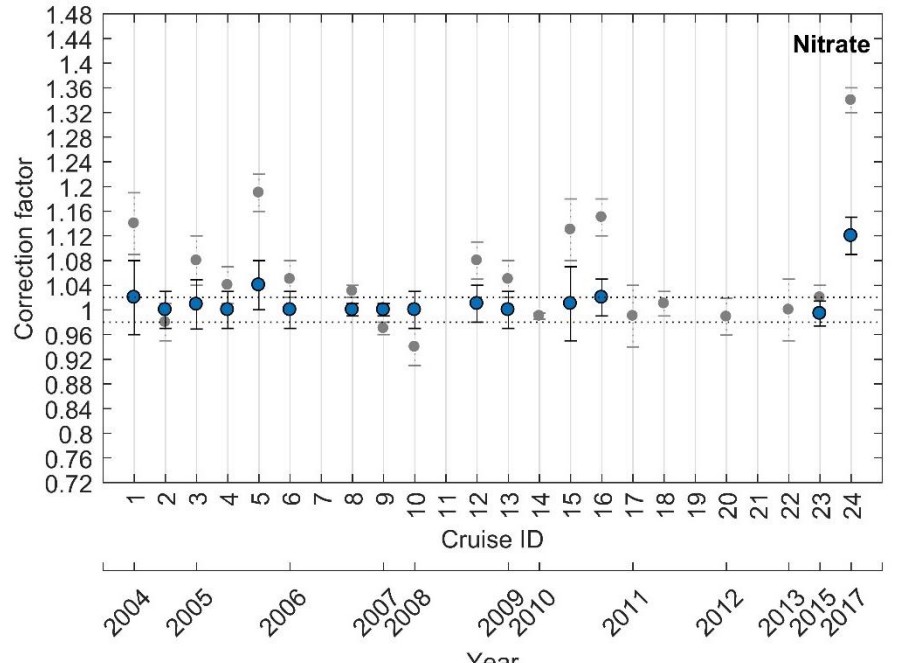
















**Figure 6**

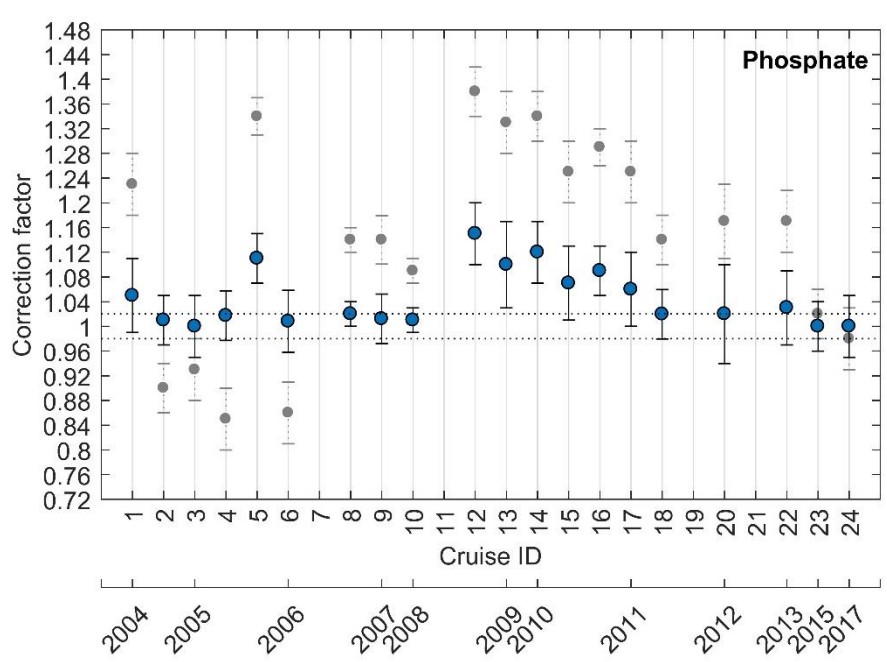







end










**Figure 7**

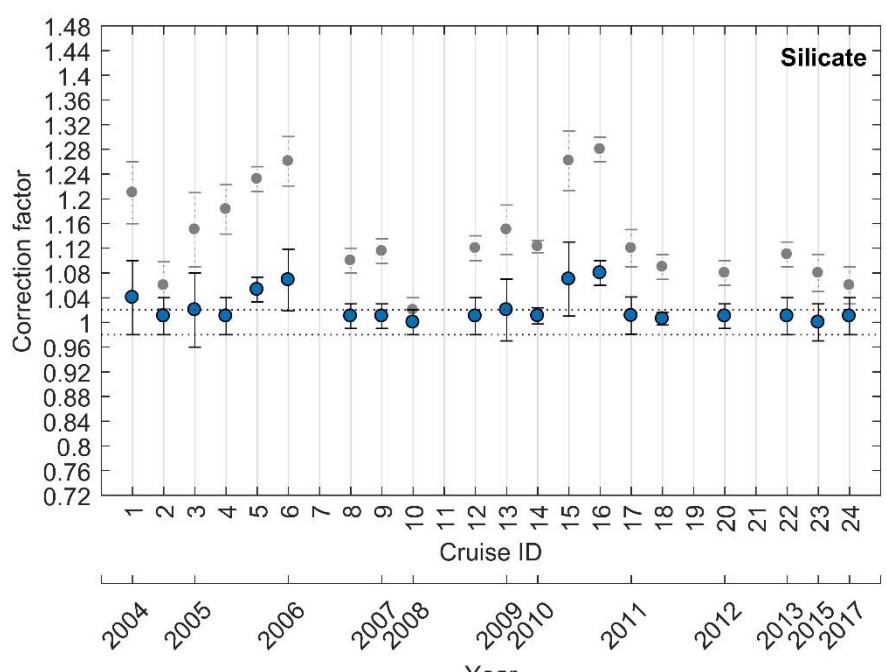










**Figure 8**

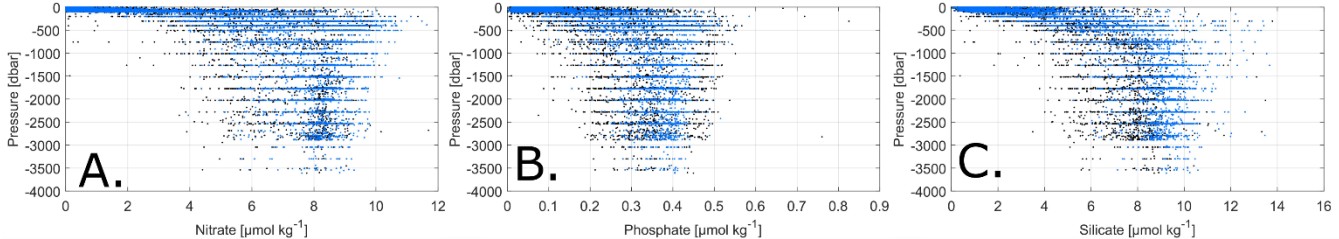

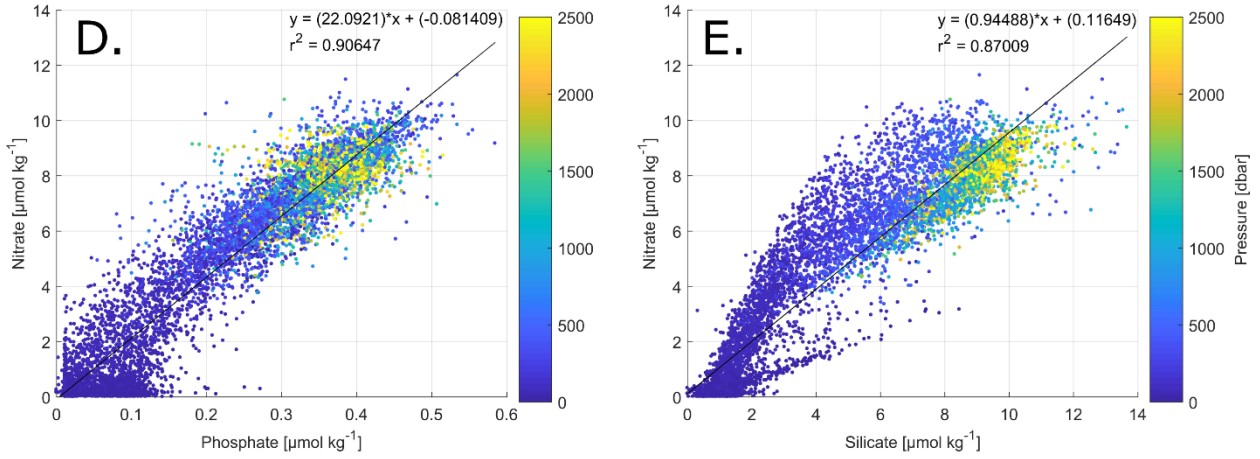















**Figure 9**

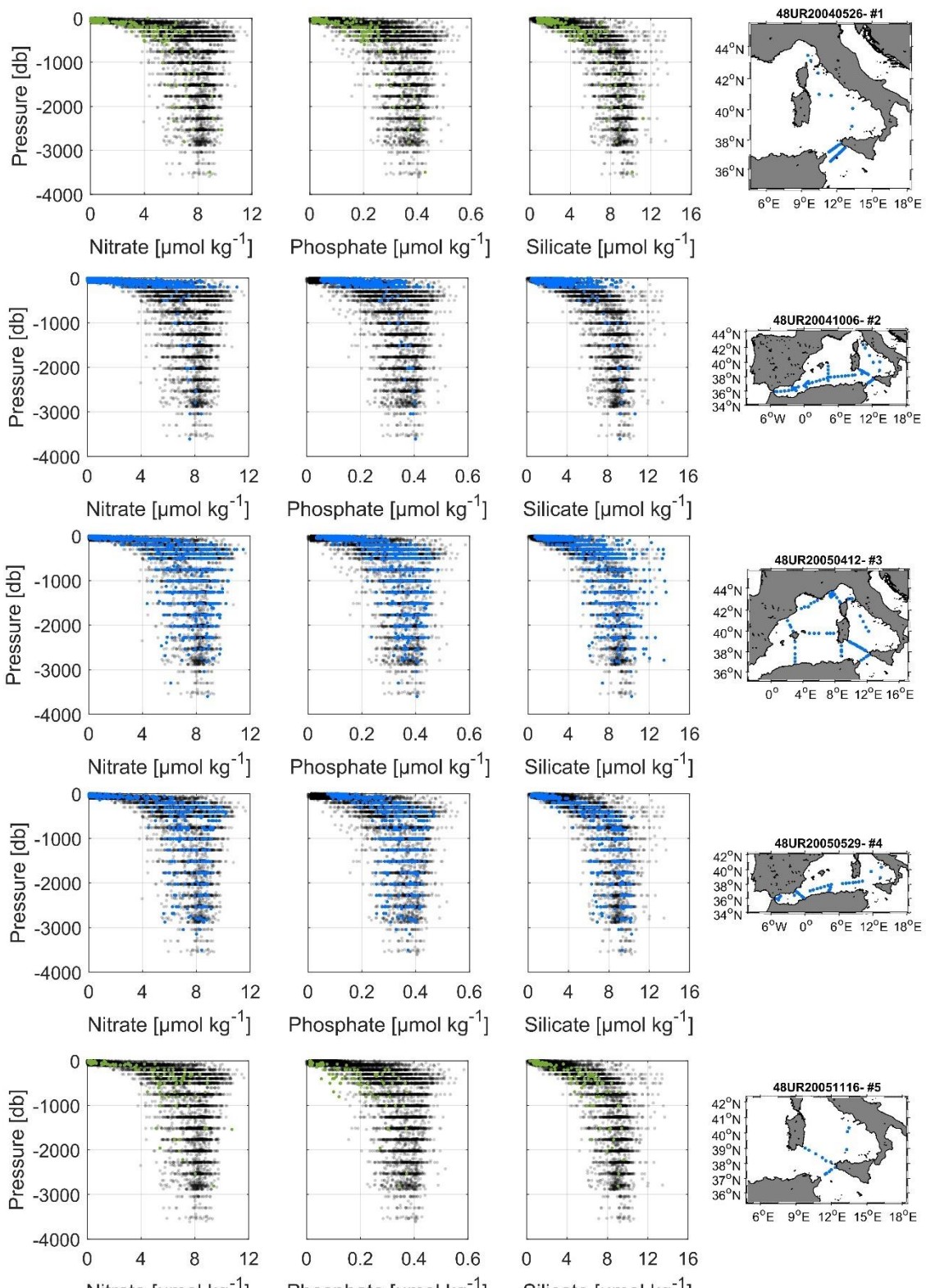

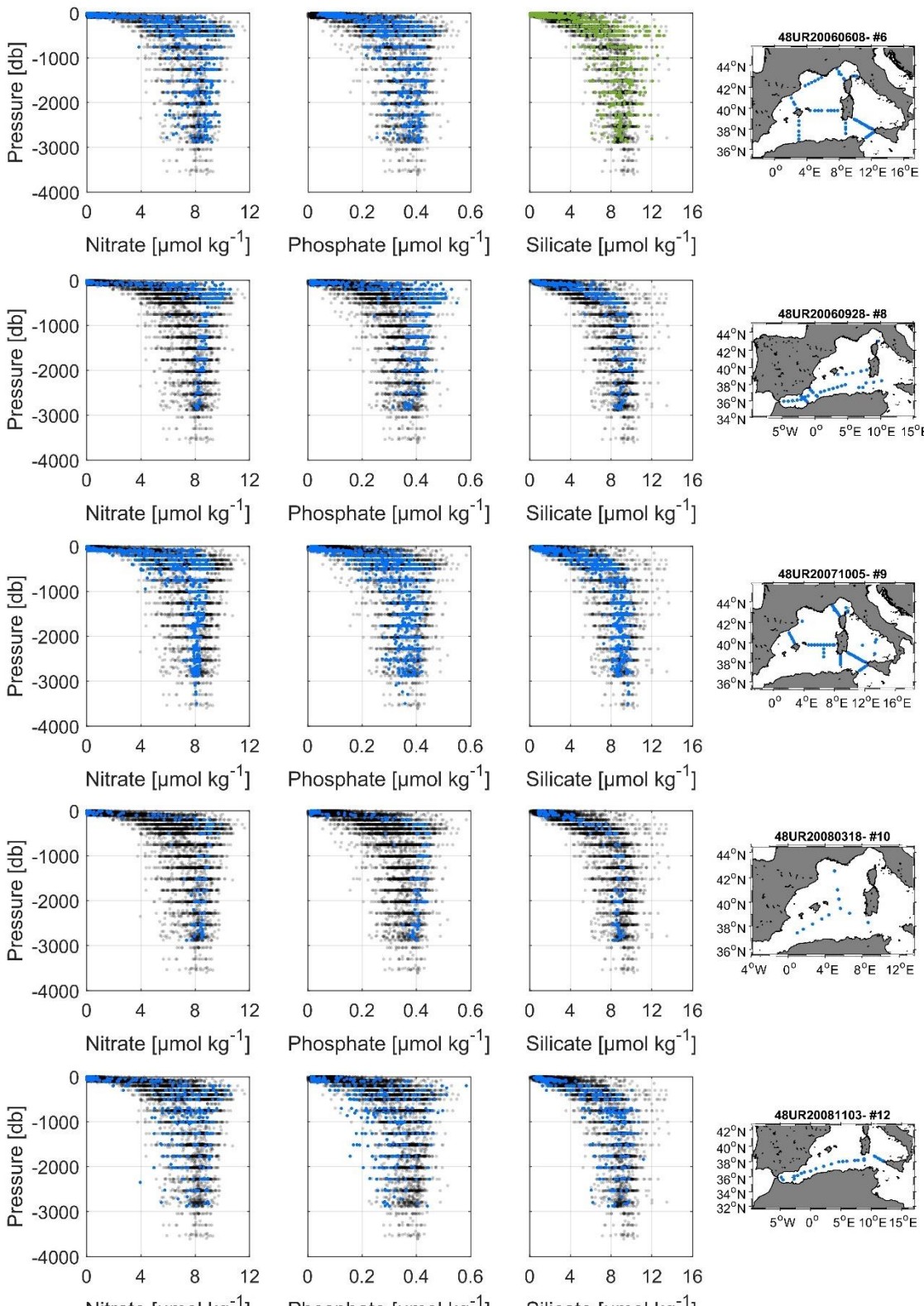


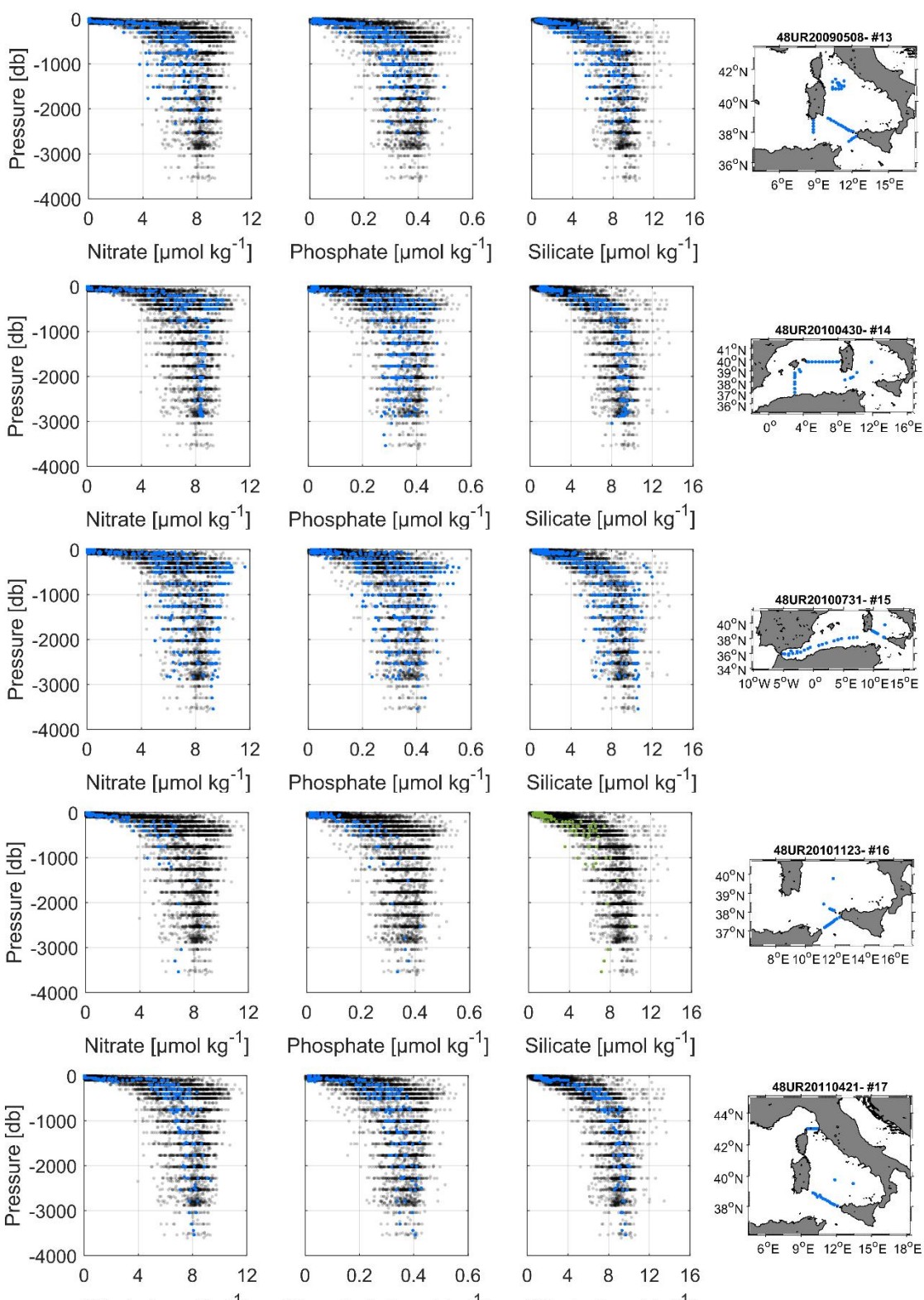

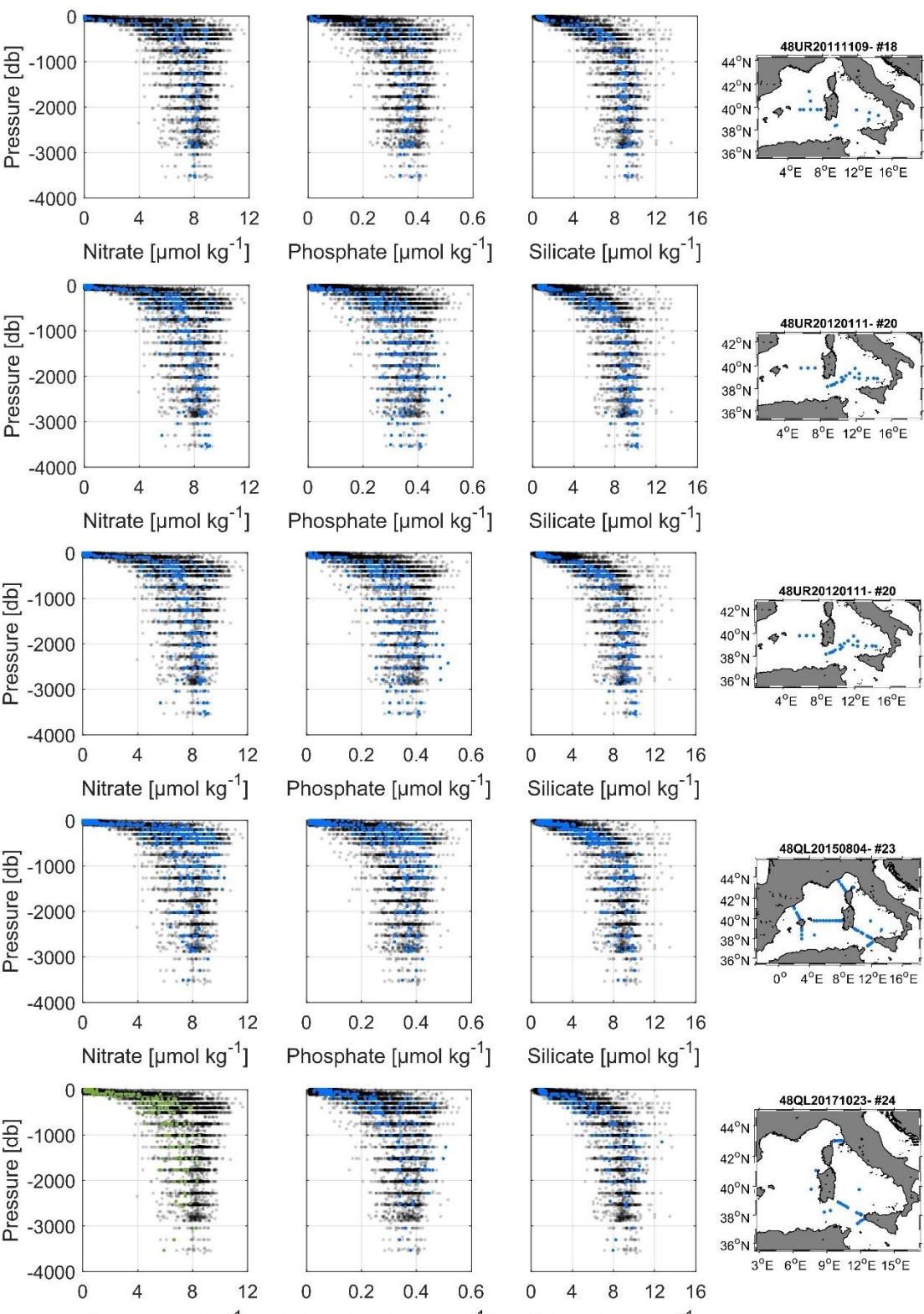



**Figure 10**

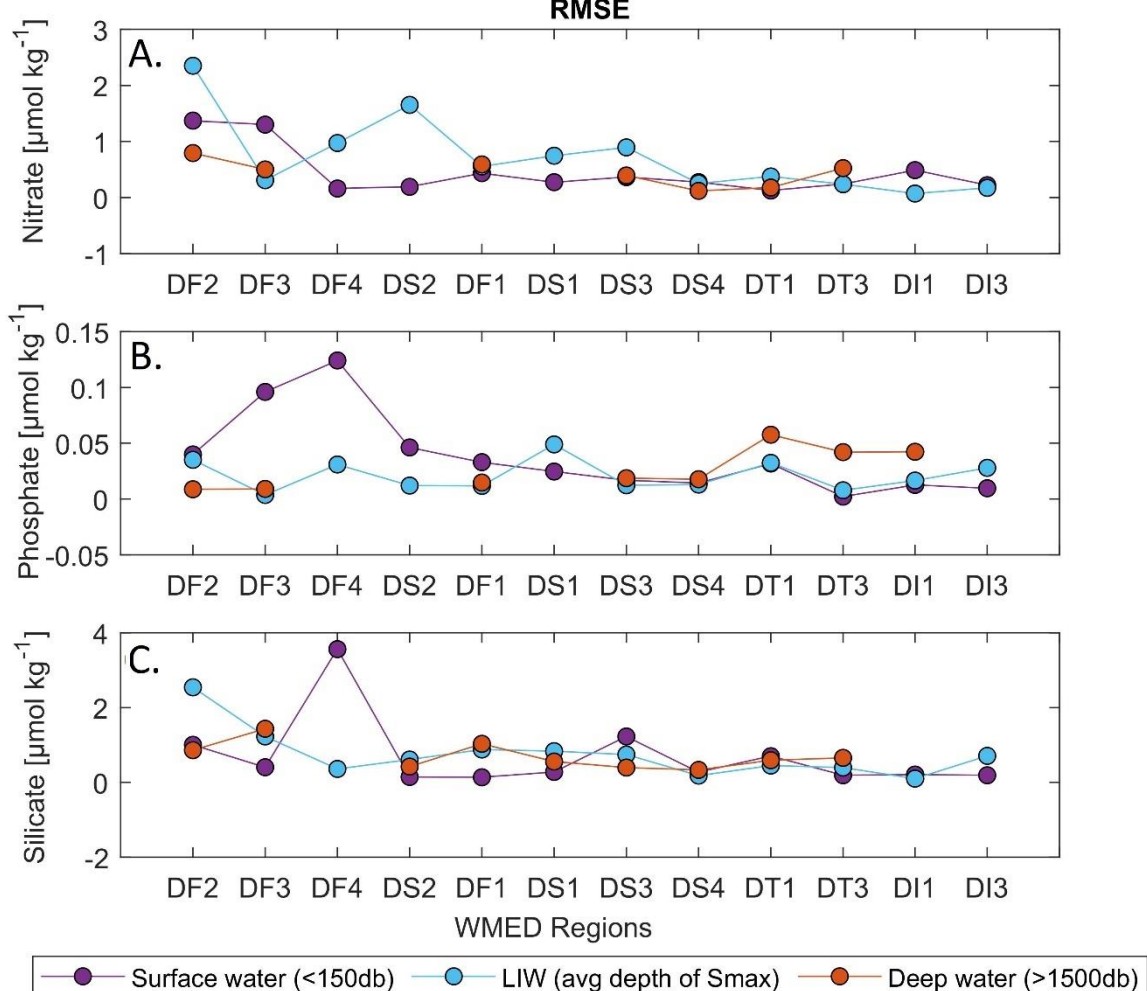










**Table 1a**

| Cruise ID (#) | Common Name | EXPOCODE | Research vessel (RV) | Date Start/End | Stations | Samples Nitrate | Samples Phosphate | Samples Silicate | Maximum bottom depth (m) | Chief scientist |
|---|---|---|---|---|---|---|---|---|---|---|
| 1 | TRENDS2004/MEDGOOS8leg2 | 48UR20040526 | Urania | 26 MAY - 14 JUN 2004 | 36 | 255 | 253 | 255 | 3499 | M. Borghini |
| 2 | MEDGOOS9 | 48UR20041006 | Urania | 6 - 25 OCT 2004 | 68 | 627 | 626 | 627 | 3610 | M. Borghini |
| 3 | MEDOCC05/MFSTEP2 | 48UR20050412 | Urania | 12 APR - 16 MAY 2005 | 68 | 828 | 828 | 828 | 3598 | M. Borghini |
| 4 | MEDGOOS10 | 48UR20050529 | Urania | 29 MAY - 10 JUN 2005 | 36 | 577 | 577 | 577 | 3505 | A. Perilli |
| 5 | MEDGOOS11 | 48UR20051116 | Urania | 16 NOV - 3 DEC 2005 | 14 | 143 | 143 | 143 | 2810 | A. Perilli, M. Borghini, M. Dibitetto |
| 6 | MEDOCC06 | 48UR20060608 | Urania | 8 JUN - 3 JUL 2006 | 66 | 787 | 785 | 787 | 2881 | M. Borghini |
| 7 | SIRENA06 | 06A420060720 | NRV Alliance | 20 JUL - 6 AUG 2006 | 35 | 208 | 208 | 209 | 1854 | J. Haun |
| 8 | MEDGOOS13/MEDBIO06 | 48UR20060928 | Urania | 28 SEP - 8 NOV 2006 | 37 | 519 | 520 | 520 | 2862 | A. Ribotti |
| 9 | MEDOCC07 | 48UR20071005 | Urania | 5 - 29 OCT 2007 | 71 | 977 | 977 | 979 | 3497 | A. Perilli, M. Borghini A. Ribotti |
| 10 | SESAMEIt4 | 48UR20080318 | Urania | 18 MAR - 7 APR 2008 | 11 | 164 | 164 | 164 | 2882 | C. Santinelli |
| 11 | SESAMEIT5 | 48UR20080905 | Urania | 5 - 16 SEP 2008 | 12 | 74 | 74 | 74 | 536 | S. Sparnocchia, G.P. Gasparini, M. Borghini |
| 12 | MEDCO08 | 48UR20081103 | Urania | 3 - 24 NOV 2008 | 24 | 342 | 350 | 348 | 2880 | A. Ribotti |
| 13 | TYRRMOUNTS | 48UR20090508 | Urania | 8 MAY - 3 JUN 2009 | 41 | 430 | 441 | 440 | 2559 | G.P. Gasparini |
| 14 | BIOFUN010 | 48UR20100430 | Urania | 30 APR - 17 MAY 2010 | 26 | 405 | 405 | 405 | 3540 | E. Manini, S. Aliani |
| 15 | VENUS1 | 48UR20100731 | Urania | 31 JUL - 25 AUG 2010 | 32 | 431 | 432 | 428 | 3544 | G.P. Gasparini, M. Borghini |
| 16 | BONSIC2010 | 48UR20101123 | Urania | 23 NOV - 9 DEC 2010 | 18 | 144 | 143 | 143 | 3540 | A. Ribotti |
| 17 | EUROFLEET11 | 48UR20110421 | Urania | 21 APR - 8 MAY 2011 | 28 | 277 | 275 | 277 | 3540 | G.P. Gasparini, M. Borghini |
| 18 | BONIFACIO2011 | 48UR20111109 | Urania | 9 - 23 NOV 2011 | 13 | 180 | 180 | 181 | 3541 | A. Ribotti, G. La Spada, M. Borghini |
| 19 | TOSCA2011 | 48MG20111210 | Maria Grazia | 10 - 20 DEC 2011 | 21 | 310 | 310 | 309 | 2728 | M. Borghini |
| 20 | ICHNUSSA12 | 48UR20120111 | Urania | 11 - 27 JAN 2012 | 21 | 353 | 352 | 323 | 3551 | A. Ribotti |
| 21 | EUROFLEET2012 | 48UR20121108 | Urania | 8 - 26 NOV 2012 | 53 | 429 | 434 | 434 | 2633 | M. Borghini |
| 22 | ICHNUSSA13 | 48UR20131015 | Urania | 15 - 29 OCT 2013 | 37 | 405 | 404 | 405 | 3540 | A. Ribotti |
| 23 | OCEANCERTAIN15 | 48QL20150804 | Minerva Uno | 4 - 29 AUG 2015 | 71 | 531 | 531 | 531 | 3513 | J. Chiggiato |
| 24 | ICHNUSSA17/INFRAOCE17 | 48QL20171023 | Minerva Uno | 23 OCT- 28 NOV 2017 | 31 | 251 | 254 | 254 | 3536 | A. Ribotti, S. Sparnocchia, M. Borghini |

**Table 1b**

| Cruise ID (#) | Expedition original Name | PIs/ Chief scientist | Specific link* (accessed June 2020) |
|---|---|---|---|
| 1 | TRENDS2004/ MEDGOOS8leg2 | M. Borghini | https://isramar.ocean.org.il/perseus_data/CruiseInfo.aspx?criuseid=5821 https://isramar.ocean.org.il/perseus_data/CruiseInfo.aspx?criuseid=4935 |
| 2 | MEDGOOS9 | M. Borghini | Report submission in progress https://isramar.ocean.org.il/perseus_data/CruiseInfo.aspx?criuseid=5823 https://doi.org/10.17882/70340 |
| 3 | MEDOCC05/ MFSTEP2 | M. Borghini | http://ricerca.ismar.cnr.it/CRUISE_REPORTS/2005/URANIA_MEDOCC05.pdf https://isramar.ocean.org.il/perseus_data/CruiseInfo.aspx?criuseid=4936 |
| 4 | MEDGOOS10 | A. Perilli | http://www.seaforecast.cnr.it/it/observation_it.htm https://doi.org/10.17882/70340 |
| 5 | MEDGOOS11 | A. Perilli, M. Borghini, M. Dibitetto | http://ricerca.ismar.cnr.it/CRUISE_REPORTS/2005/URANIA_MEDGOOS11_05_REP.pdf https://doi.org/10.17882/70340 |
| 6 | MEDOCC06 | M. Borghini | http://www.seaforecast.cnr.it/reports/Medocc06CR.pdf https://seadata.bsh.de/Cgi-csr/retrieve_sdn2/viewReport.pl?csrref=20106010 |
| 7 | SIRENA06 | J. Haun | Report submission in progress |
| 8 | MEDGOOS13/ MEDBIO06 | A. Ribotti | http://www.seaforecast.cnr.it/reports/Mebio06-Medg13_CR.pdf https://doi.org/10.17882/70340 |
| 9 | MEDOCC07 | A. Perilli, M. Borghini, A. Ribotti | http://www.seaforecast.cnr.it/reports/Medocc07-MedCo07_Rapp.pdf https://isramar.ocean.org.il/perseus_data/CruiseInfo.aspx?criuseid=5146 |
| 10 | SESAMEIt4 | C. Santinelli | https://isramar.ocean.org.il/perseus_data/CruiseInfo.aspx?criuseid=5148 https://emodnet-chemistry.maris.nl/search/details.php?step=0012004~0022017~0153~057l04001~058tdin,ntra,phos,slca~00445~0056~00617~00734~0541&count=3592&page=1000&sort=0&header=no |
| 11 | SESAMEIT5 | S. Sparnocchia, G.P. Gasparini, M. Borghini | https://isramar.ocean.org.il/perseus_data/CruiseInfo.aspx?criuseid=5147 |
| 12 | MEDCO08 | A. Ribotti | http://www.seaforecast.cnr.it/reports/MedCO08_Rapp.pdf |
| 13 | TYRRMOUNTS | G.P. Gasparini | Report submission in progress |
| 14 | BIOFUN010 | E. Manini, S. Aliani | http://www.ismar.cnr.it/products/reports-campagne/2010-2019 |
| 15 | VENUS1 | G.P. Gasparini, M. Borghini | Report submission in progress |
| 16 | BONSIC2010 | A. Ribotti | http://www.seaforecast.cnr.it/reports/Bonifacio2010Sic_Rapp.pdf |
| 17 | EUROFLEET11 | G.P. Gasparini, M. Borghini | Report submission in progress |
| 18 | BONIFACIO2011 | A. Ribotti, G. La Spada, M. Borghini | http://www.seaforecast.cnr.it/reports/Bonifacio2011_Rapp.pdf |
| 19 | TOSCA2011 | M. Borghini | Report submission in progress |
| 20 | ICHNUSSA12 | A. Ribotti | http://www.seaforecast.cnr.it/reports/Ichnussa2012_Rapp.pdf |
| 21 | EUROFLEET2012 | M. Borghini | Report submission in progress |
| 22 | ICHNUSSA13 | A. Ribotti | http://www.seaforecast.cnr.it/reports/Ichnussa2013_Rapp.pdf |
| 23 | OCEANCERTAIN15 | J. Chiggiato | https://doi.org/10.1594/PANGAEA.911046 |
| 24 | ICHNUSSA17/ INFRAOCE17 | A. Ribotti, S. Sparnocchia, M. Borghini | Report submission in progress |

\* The specific links are subjected to updates.

**Table 2**

| Common name | EXPOCODE | Date Start/End | Stations | Nitrate Sample | Phosphate Sample | Silicate Sample | Source | Nutrient PI | Chief scientist |
|---|---|---|---|---|---|---|---|---|---|
| *M51/2* | 06MT20011018 | 18 OCT - 11 NOV 2001 | 6 | 79 | 79 | 82 | GLODAPv2 | B. Schneider | W. Roether |
| *TRANSMED_LEGII* | 48UR20070528 | 28 MAY- 12 JUN 2007 | 4 | 78 | 77 | 78 | CARIMED (not yet available) | S. Cozzi, V. Ibello | M. Azzaro |
| *M84/3* | 06MT20110405 | 5 - 28 APR 2011 | 20 | 339 | 343 | - | GLODAPv2 | G. Civitarese | T. Tanhua |
| *HOTMIX* | 29AH20140426 | 26 APR- 31 MAY 2014 | 18 | 144 | 140 | 144 | CARIMED (not yet available) | XA Álvarez-Salgado | J. Aristegui |
| *TALPro-2016* | 29AJ20160818 | 18 - 28 AUG 2016 | 42 | 293 | 293 | 293 | MedSHIP programme | L. Coppola | L. Jullion, K. Schroeder |

**Table 3**

| WOCE flag value | Interpretation in original dataset | Interpretation in adjusted product |
|---|---|---|
| 2 | Acceptable/ measured | Adjusted and acceptable |
| 3 | Questionable/not used | Adjusted and recommended questionable |
| 9 | not measured/no data | - |

**Table 4**

| Cruise ID | EXPOCODE/ Region | Regional Avg Nitrate (µmol | std Nitrate | Regional Avg Phosphate | std Phosphate( | Regional Avg Silicate (µmol | std Silicate | # samples | Avg storage (in |
|---|---|---|---|---|---|---|---|---|---|

| | | kg⁻¹) | (μmol kg⁻¹) | (μmol kg⁻¹) | μmol kg⁻¹ | kg⁻¹) | (μmol kg⁻¹) | | days) |
|---|---|---|---|---|---|---|---|---|---|
| 1 | 48UR20040526/ | | **1.25** | | **0.062** | | **1.64** | 21 | 131 |
| | *DT1-Tyrrhenian North* | 6.07 | 1.32 | 0.26 | 0.065 | 6.92 | 1.83 | 16 | |
| | *DT3-Tyrrhenian South* | 7.03 | 0.51 | 0.31 | 0.02 | 7.66 | 0.53 | 5 | |
| 2 | 48UR20041006/ | | **0.59** | | **0.029** | | **0.81** | 21 | 251 |
| | *DT1-Tyrrhenian North* | 7.68 | 0.53 | 0.41 | 0.031 | 8.74 | 0.75 | 15 | |
| | *DT3-Tyrrhenian South* | 8.17 | 0.60 | 0.41 | 0.025 | 9.31 | 0.87 | 6 | |
| 3 | 48UR20050412/ | | **1.15** | | **0.050** | | **1.41** | 233 | 135 |
| | *DF2-Gulf of Lion* | 7.89 | 0.98 | 0.40 | 0.044 | 8.17 | 1.065 | 24 | |
| | *DF3-Liguro-Provençal* | 7.45 | 1.08 | 0.41 | 0.05 | 7.72 | 1.10 | 66 | |
| | *DS2-Balearic Sea* | 7.44 | 1.14 | 0.40 | 0.039 | 7.68 | 1.47 | 21 | |
| | *DF1-Algero-Provençal* | 7.87 | 1.16 | 0.41 | 0.043 | 8.88 | 1.96 | 42 | |
| | *DS3-Algerian West* | 7.7 | 0.816 | 0.39 | 0.048 | 8.14 | 0.941 | 23 | |
| | *DT1-Tyrrhenian North* | 6.57 | 1.065 | 0.36 | 0.047 | 7.41 | 1.15 | 21 | |
| | *DT3-Tyrrhenian South* | 6.52 | 1.12 | 0.36 | 0.05 | 7.56 | 1.42 | 22 | |
| | *DI1-Sardinia Channel* | 7.22 | 1.065 | 0.40 | 0.04 | 8.08 | 1.11 | 14 | |
| 4 | 48UR20050529/ | | **1.13** | | **0.057** | | **1.08** | 205 | 314 |
| | *DS1-Alboran Sea* | 6.4 | 1.15 | 0.38 | 0.041 | 6.26 | 1.02 | 32 | |
| | *DS3-Algerian West* | 7.6 | 1.13 | 0.41 | 0.06 | 7.33 | 0.99 | 73 | |
| | *DS4-Algerian East* | 7.48 | 1.13 | 0.41 | 0.06 | 7.50 | 1.23 | 47 | |
| | *DT1-Tyrrhenian North* | 7.24 | 0.44 | 0.42 | 0.03 | 7.91 | 0.56 | 16 | |
| | *DT3-Tyrrhenian South* | 7.70 | 0.38 | 0.41 | 0.03 | 7.55 | 0.36 | 14 | |
| | *DI1-Sardinia Channel* | 7.58 | 1.08 | 0.43 | 0.049 | 7.42 | 0.82 | 23 | |
| 5 | 48UR20051116/ | | **1.35** | | **0.078** | | **0.98** | 16 | 738 |
| | *DT1-Tyrrhenian North* | 5.68 | 1.26 | 0.19 | 0.08 | 6.30 | 0.92 | 10 | |
| | *DT3-Tyrrhenian South* | 6.71 | 1.51 | 0.20 | 0.06 | 6.86 | 1.065 | 5 | |
| | *DI1-Sardinia Channel* | 6.29 | 0 | 0.26 | 0 | 7.53 | 0 | 1 | |
| 6 | 48UR20060608/ | | **1.16** | | **0.054** | | **1.47** | 221 | 27 |
| | *DF2-Gulf of Lion* | 7.69 | 1.02 | 0.42 | 0.04 | 7.089 | 1.04 | 27 | |
| | *DF3-Liguro-Provençal* | 8.08 | 0.78 | 0.43 | 0.04 | 7.41 | 1.21 | 35 | |
| | *DS2-Balearic Sea* | 8.06 | 0.9 | 0.43 | 0.03 | 7.07 | 1.18 | 30 | |
| | *DF1-Algero-Provençal* | 7.97 | 1.16 | 0.44 | 0.05 | 7.34 | 1.32 | 61 | |
| | *DS3-Algerian West* | 8.39 | 0.9 | 0.42 | 0.03 | 8.5 | 2 | 28 | |
| | *DT3-Tyrrhenian South* | 6.39 | 1.28 | 0.36 | 0.06 | 6.86 | 1.7 | 26 | |
| | *DI1-Sardinia Channel* | 8.04 | 0.85 | 0.43 | 0.04 | 7.77 | 1.25 | 14 | |
| 7 | 06A420060720 | | - | | - | | - | - | 1367 |
| 8 | 48UR20060928/ | | **0.71** | | **0.036** | | **0.76** | 179 | 606 |
| | *DS2-Balearic Sea* | 7.97 | 0.17 | 0.33 | 0.017 | 7.84 | 0.27 | 4 | |
| | *DF1-Algero-Provençal* | 8.17 | 0.22 | 0.33 | 0.026 | 8.11 | 0.3 | 22 | |
| | *DS1-Alboran Sea* | 8.2 | 0.14 | 0.35 | 0.02 | 8.59 | 0.35 | 47 | |
| | *DS3-Algerian West* | 7.93 | 0.89 | 0.33 | 0.03 | 8.09 | 0.91 | 70 | |
| | *DS4-Algerian East* | 7.98 | 0.68 | 0.34 | 0.04 | 8.01 | 0.7 | 28 | |
| | *DT3-Tyrrhenian South* | 6.2 | 1.51 | 0.28 | 0.04 | 6.71 | 1.45 | 3 | |
| | *DI1-Sardinia Channel* | 7.66 | 0.6 | 0.28 | 0.02 | 8.00 | 0.49 | 5 | |
| 9 | 48UR20071005/ | | **0.89** | | **0.040** | | **0.86** | 302 | 751 |
| | *DF2-Gulf of Lion* | 8.41 | 0.08 | 0.31 | 0.01 | 7.43 | 0.02 | 4 | |
| | *DF3-Liguro-Provençal* | 8.17 | 1.08 | 0.31 | 0.03 | 7.64 | 1.08 | 81 | |
| | *DS2-Balearic Sea* | 8.17 | 0.43 | 0.31 | 0.02 | 7.58 | 0.39 | 29 | |
| | *DF1-Algero-Provençal* | 8.33 | 0.6 | 0.32 | 0.03 | 7.79 | 0.69 | 82 | |
| | *DS4-Algerian East* | 8.41 | 0.2 | 0.33 | 0.018 | 7.90 | 0.26 | 19 | |
| | *DT1-Tyrrhenian North* | 7.83 | 0.41 | 0.28 | 0.03 | 8.26 | 0.55 | 26 | |
| | *DT3-Tyrrhenian South* | 7.49 | 1.22 | 0.28 | 0.05 | 7.71 | 1.26 | 38 | |
| | *DI1-Sardinia Channel* | 7.92 | 1.05 | 0.33 | 0.02 | 8.26 | 0.41 | 23 | |
| 10 | 48UR20080318/ | | **0.51** | | **0.026** | | **0.34** | 66 | 31 |
| | *DF2-Gulf of Lion* | 8.54 | 0.6 | 0.35 | 0.03 | 8.62 | 0.43 | 5 | |
| | *DS2-Balearic Sea* | 9.12 | 0.18 | 0.38 | 0.01 | 8.40 | 0.21 | 9 | |
| | *DF1-Algero-Provençal* | 9.02 | 0.36 | 0.38 | 0.03 | 8.65 | 0.25 | 15 | |
| | *DS3-Algerian West* | 8.93 | 0.46 | 0.36 | 0.01 | 8.69 | 0.35 | 20 | |
| | *DS4-Algerian East* | 8.43 | 0.25 | 0.38 | 0.02 | 8.32 | 0.22 | 10 | |
| | *DI1-Sardinia Channel* | 7.62 | 0.6 | 0.34 | 0.03 | 8.49 | 0.36 | 3 | |
| 11* | 48UR20080905 | | - | | - | | - | - | 211 |
| 12 | 48UR20081103/ | | **1.11** | | **0.077** | | **0.10** | 110 | 536 |
| | *DS1-Alboran Sea* | 6.4 | 1.21 | 0.21 | 0.06 | 7.20 | 1.43 | 26 | |
| | *DS3-Algerian West* | 7.58 | 0.9 | 0.27 | 0.1 | 7.89 | 0.9 | 30 | |
| | *DS4-Algerian East* | 7.15 | 1.04 | 0.23 | 0.04 | 7.38 | 0.9 | 35 | |
| | *DT3-Tyrrhenian South* | 7.44 | 0.5 | 0.22 | 0.05 | 8.28 | 0.4 | 10 | |
| | *DI1-Sardinia Channel* | 7.40 | 1.23 | 0.17 | 0.04 | 8.09 | 0.45 | 9 | |
| 13 | 48UR20090508/ | | **1.41** | | **0.051** | | **1.42** | 88 | 164 |
| | *DT1-Tyrrhenian North* | 5.95 | 1.55 | 0.24 | 0.05 | 6.28 | 1.58 | 46 | |
| | *DT3-Tyrrhenian South* | 6.76 | 0.77 | 0.24 | 0.03 | 7.37 | 0.77 | 29 | |
| | *DI1-Sardinia Channel* | 7.62 | 1.1 | 0.28 | 0.05 | 7.76 | 0.9 | 13 | |
| 14 | 48UR20100430/ | | **1.06** | | **0.036** | | **1.03** | 159 | 213 |
| | *DS2-Balearic Sea* | 7.66 | 1.6 | 0.25 | 0.03 | 7.38 | 1.75 | 33 | |
| | *DF1-Algero-Provençal* | 8.43 | 0.29 | 0.26 | 0.03 | 8.06 | 0.31 | 61 | |

| # | EXPOCODE / Region | | | | | | | | |
|---|---|---|---|---|---|---|---|---|---|
| | *DS3-Algerian West* | 8.5 | 0.14 | 0.26 | 0.03 | 8.25 | 0.3 | 26 | |
| | *DT1-Tyrrhenian North* | 6.88 | 0.8 | 0.23 | 0.022 | 7.17 | 0.77 | 11 | |
| | *DT3-Tyrrhenian South* | 6.38 | 1.35 | 0.22 | 0.01 | 6.76 | 1.56 | 7 | |
| | *DI1-Sardinia Channel* | 7.71 | 0.87 | 0.23 | 0.02 | 7.80 | 0.74 | 21 | |
| 15 | 48UR20100731/ | | **1.34** | | **0.053** | | **0.14** | 149 | 213 |
| | *DS1-Alboran Sea* | 7.30 | 1.18 | 0.29 | 0.05 | 7.21 | 1.11 | 25 | |
| | *DS3-Algerian West* | 7.67 | 1.15 | 0.28 | 0.045 | 7.24 | 1.16 | 54 | |
| | *DS4-Algerian East* | 7.38 | 0.89 | 0.29 | 0.03 | 7.00 | 0.78 | 29 | |
| | *DT1-Tyrrhenian North* | 7.66 | 0.96 | 0.29 | 0.05 | 7.89 | 1.07 | 10 | |
| | *DT3-Tyrrhenian South* | 5.4 | 0.67 | 0.22 | 0.01 | 5.52 | 1.56 | 30 | |
| | *DI1-Sardinia Channel* | 4.92 | 0 | 0.20 | 0 | 5.55 | 0 | 1 | |
| 16 | 48UR20101123/ | | **1.02** | | **0.045** | | **1.02** | 14 | 170 |
| | DT1-Tyrrhenian North | 6.34 | 0.87 | 0.27 | 0.02 | 6.12 | 0.87 | 8 | |
| | DT3-Tyrrhenian South | 5.43 | 1.02 | 0.22 | 0.04 | 5.08 | 0.9 | 6 | |
| 17 | 48UR20110421/ | | **0.62** | | **0.029** | | **0.52** | 56 | 160 |
| | DT1-Tyrrhenian North | 7.77 | 0.45 | 0.28 | 0.02 | 8.11 | 0.35 | 21 | |
| | DT3-Tyrrhenian South | 7.76 | 0.7 | 0.28 | 0.03 | 8.017 | 0.6 | 35 | |
| 18 | 48UR20111109/ | | **0.68** | | **0.025** | | **0.70** | 77 | 74 |
| | DF3-Liguro-Provençal | 6.68 | 0 | 0.33 | 0 | 6.26 | 0 | 1 | |
| | DF1-Algero-Provençal | 8.17 | 0.5 | 0.32 | 0.01 | 8.16 | 0.66 | 43 | |
| | DT1-Tyrrhenian North | 7.26 | 0.93 | 0.29 | 0.02 | 8.15 | 1.03 | 12 | |
| | DT3-Tyrrhenian South | 7.61 | 0.37 | 0.30 | 0.02 | 8.18 | 0.35 | 11 | |
| | DI1-Sardinia Channel | 7.64 | 0.45 | 0.29 | 0.01 | 8.08 | 0.41 | 10 | |
| 19* | 48MG20111210 | | - | | - | | - | - | 38 |
| 20 | 48UR20120111/ | | **0.97** | | **0.051** | | **0.26** | 152 | 317 |
| | DF1-Algero-Provençal | 8.45 | 0.49 | 0.31 | 0.039 | 7.91 | 0.53 | 23 | |
| | DT1-Tyrrhenian North | 7.67 | 0.83 | 0.27 | 0.02 | 8.29 | 0.8 | 30 | |
| | DT3-Tyrrhenian South | 7.65 | 1.06 | 0.31 | 0.06 | 8.03 | 1.26 | 69 | |
| | DI1-Sardinia Channel | 7.65 | 0.96 | 0.31 | 0.03 | 7.86 | 0.78 | 30 | |
| 21* | 48UR20121108 | | - | | - | | - | - | 72 |
| 22 | 48UR20131015/ | | **1.03** | | **0.043** | | **0.79** | 98 | 76 |
| | *DF1-Algero-Provençal* | 8.54 | 0.64 | 0.33 | 0.02 | 7.96 | 0.38 | 36 | |
| | *DS4-Algerian East* | 7.67 | 1.28 | 0.27 | 0.04 | 6.82 | 1.07 | 8 | |
| | *DT1-Tyrrhenian North* | 6.47 | 0.83 | 0.24 | 0.025 | 7.12 | 0.84 | 10 | |
| | *DT3-Tyrrhenian South* | 7.81 | 0.71 | 0.30 | 0.03 | 8.09 | 0.65 | 28 | |
| | *DI1-Sardinia Channel* | 7.32 | 0.99 | 0.27 | 0.02 | 7.47 | 0.89 | 16 | |
| 23 | 48QL20150804/ | | **0.84** | | **0.038** | | **0.85** | 94 | 30 |
| | *DF3-Liguro-Provençal* | 8.51 | 0.96 | 0.39 | 0.03 | 8.06 | 0.85 | 23 | |
| | *DS2-Balearic Sea* | 7.75 | 0.66 | 0.36 | 0.02 | 7.86 | 0.81 | 20 | |
| | *DF1-Algero-Provençal* | 7.9 | 0.59 | 0.37 | 0.03 | 8.34 | 0.68 | 23 | |
| | *DS3-Algerian West* | 7.84 | 0.67 | 0.36 | 0.02 | 7.75 | 0.68 | 6 | |
| | *DT1-Tyrrhenian North* | 7.92 | 0.61 | 0.37 | 0.02 | 8.75 | 0.4 | 8 | |
| | *DT3-Tyrrhenian South* | 7.23 | 0.75 | 0.34 | 0.025 | 8.2 | 0.94 | 13 | |
| | *DI1-Sardinia Channel* | 6.30 | 0 | 0.25 | 0 | 5.36 | 0 | 1 | |
| 24 | 48QL20171023/ | | **0.68** | | **0.055** | | **1.24** | 55 | 30 |
| | *DF3-Liguro-Provençal* | 6.63 | 0.41 | 0.40 | 0.05 | 10.76 | 1.07 | 3 | |
| | *DF1-Algero-Provençal* | 5.14 | 0.7 | 0.43 | 0.02 | 7.94 | 1.19 | 6 | |
| | *DT1-Tyrrhenian North* | 4.98 | 0.58 | 0.36 | 0.02 | 8.10 | 0.87 | 9 | |
| | *DT3-Tyrrhenian South* | 5.43 | 0.5 | 0.36 | 0.04 | 9.03 | 0.87 | 26 | |
| | *DI1-Sardinia Channel* | 5.16 | 0.76 | 0.41 | 0.07 | 7.58 | 1.17 | 11 | |

(*) cruise not included in the 2$^{nd}$QC (Section 4.)

in bold: the overall standard deviation by cruise; in normal font: regional standard deviation by cruise

**Table 5**

| Cruise ID | EXPOCODE | Nitrate (x) | Phosphate (x) | Silicate (x) |
|---|---|---|---|---|
| 1 | 48UR20040526 | 1.14 | 1.23 | 1.21 |
| 2 | 48UR20041006 | 0.98 | 0.9 | 1.06 |
| 3 | 48UR20050412 | 1.08 | 0.93 | 1.15 |
| 4 | 48UR20050529 | 1.04 | 0.85 | 1.183 |
| 5 | 48UR20051116 | 1.19 | 1.34 | 1.232 |

| | | | | |
|---|---|---|---|---|
| 6 | 48UR20060608 | 1.05 | 0.86 | 1.261 |
| 7 | 06A420060720* | - | - | - |
| 8 | 48UR20060928 | 1.03 | 1.14 | 1.1 |
| 9 | 48UR20071005 | 0.97 | 1.14 | 1.115 |
| 10 | 48UR20080318 | 0.94 | 1.09 | 1.02 |
| 11 | 48UR20080905* | - | - | - |
| 12 | 48UR20081103 | 1.08 | 1.38 | 1.12 |
| 13 | 48UR20090508 | 1.05 | 1.33 | 1.15 |
| 14 | 48UR20100430 | NA | 1.34 | 1.123 |
| 15 | 48UR20100731 | 1.13 | 1.25 | 1.262 |
| 16 | 48UR20101123 | 1.15 | 1.29 | 1.28 |
| 17 | 48UR20110421 | NA | 1.25 | 1.12 |
| 18 | 48UR20111109 | NA | 1.14 | 1.09 |
| 19 | 48MG20111210* | - | - | - |
| 20 | 48UR20120111 | NA | 1.17 | 1.08 |
| 21 | 48UR20121108* | - | - | - |
| 22 | 48UR20131015 | NA | 1.17 | 1.11 |
| 23 | 48QL20150804 | 1.02 | 1.02 | 1.08 |
| 24 | 48QL20171023 | 1.34 | 0.98 | 1.06 |

(*) cruise not included in the $2^{nd}$QC (Section 4.)

**Table 6**

| Cruise ID | EXPOCODE | Nitrate [%] | | | Phosphate[%] | | | Silicate [%] | | |
|---|---|---|---|---|---|---|---|---|---|---|
| | | *n* | *unadjusted* | *adjusted* | *n* | *unadjusted* | *adjusted* | *n* | *unadjusted* | *adjusted* |
| 1 | 48UR20040526 | 2 | 0.86 | 0.98 | 2 | 0.77 | 0.95 | 1 | 0.79 | 0.96 |
| 2 | 48UR20041006 | 2 | 1.02 | 1.00 | 2 | 1.10 | 0.99 | 1 | 0.94 | 0.99 |
| 3 | 48UR20050412 | 5 | 0.92 | 0.99 | 5 | 1.07 | 1.00 | 4 | 0.85 | 0.98 |
| 4 | 48UR20050529 | 5 | 0.96 | 1.00 | 5 | 1.15 | 0.98 | 4 | 0.82 | 0.99 |
| 5 | 48UR20051116 | 2 | 0.81 | 0.96 | 1 | 0.66 | 0.89 | 1 | 0.77 | 0.95 |
| 6 | 48UR20060608 | 5 | 0.95 | 1.00 | 5 | 1.14 | 0.99 | 4 | 0.74 | 0.93 |
| 7 | 06A420060720 | 0 | - | - | 0 | - | - | 0 | - | - |
| 8 | 48UR20060928 | 4 | 0.97 | 1.00 | 4 | 0.86 | 0.98 | 3 | 0.90 | 0.99 |
| 9 | 48UR20071005 | 5 | 1.03 | 1.00 | 5 | 0.86 | 0.98 | 4 | 0.88 | 0.99 |
| 10 | 48UR20080318 | 3 | 1.06 | 1.00 | 3 | 0.91 | 0.99 | 2 | 0.98 | 1.00 |
| 11 | 48UR20080905 | 0 | - | - | 0 | - | - | 0 | - | - |
| 12 | 48UR20081103 | 5 | 0.92 | 0.99 | 5 | 0.62 | 0.85 | 4 | 0.88 | 0.99 |
| 13 | 48UR20090508 | 3 | 0.95 | 1.00 | 3 | 0.67 | 0.90 | 2 | 0.85 | 0.98 |
| 14 | 48UR20100430 | 4 | 1.01 | NA | 4 | 0.66 | 0.88 | 3 | 0.88 | 0.99 |
| 15 | 48UR20100731 | 5 | 0.87 | 0.99 | 5 | 0.75 | 0.93 | 4 | 0.74 | 0.93 |
| 16 | 48UR20101123 | 1 | 0.85 | 0.98 | 1 | 0.71 | 0.91 | 1 | 0.72 | 0.92 |
| 17 | 48UR20110421 | 2 | 1.01 | NA | 2 | 0.75 | 0.94 | 1 | 0.88 | 0.99 |
| 18 | 48UR20111109 | 4 | 0.99 | NA | 4 | 0.86 | 0.98 | 3 | 0.91 | 0.99 |
| 19 | 48MG20111210 | 0 | - | - | 0 | - | - | 0 | - | - |
| 20 | 48UR20120111 | 4 | 1.01 | NA | 4 | 0.83 | 0.98 | 3 | 0.92 | 0.99 |
| 21 | 48UR20121108 | 0 | - | - | 0 | - | - | 0 | - | - |
| 22 | 48UR20131015 | 4 | 1.00 | NA | 4 | 0.83 | 0.97 | 3 | 0.89 | 0.99 |
| 23 | 48QL20150804 | 5 | 0.98 | 1.00 | 5 | 0.98 | 1.00 | 4 | 0.92 | 1.00 |
| 24 | 48QL20171023 | 3 | 0.66 | 0.88 | 3 | 1.02 | 1.00 | 2 | 0.94 | 0.99 |

red: data lower than reference

**Table 7**

| Region/ Water mass | Nitrate (µmol kg⁻¹) | | Phosphate (µmol kg⁻¹) | | Silicate (µmol kg⁻¹) | |
|---|---|---|---|---|---|---|
| | Avg new Product | Avg Medar | Avg new Product | Avg Medar | Avg new Product | Avg Medar |
| *DF2- Gulf of Lion* | | | | | | |
| surface water (0-150db) | 2.68±2.53(68)** | 1.7±1.1 | 0.15±0.06(68) | 0.13±0.04 | 2.91±1.33(68) | 1.72±0.64 |
| LIW core ($S_{max}$ depth range: 300-500db) | 8.49±0.18(17) | 6.13±0.32 | 0.38±0.02(17) | 0.34±0.01 | 8.67±0.69(17) | 6.12±0.61 |
| Deep water (>1500db) | 8.03±0.43(33) | 7.64±0.31 | 0.37±0.01(33) | 0.37±0.015 | 8.7±0.67(33) | 7.95±0.06 |
| *DF3- Liguro-Provençal* | | | | | | |
| surface water (0-150db) | 2.31±2.4(205) | 3.0±2.6 | 0.12±0.07(205) | 0.19±0.05 | 2.45±1.05(205) | 2.16±1.05 |
| LIW core ($S_{max}$ depth range: 300-500db) | 8.05±0.18(76) | 7.74±0.13 | 0.36±0.01(76) | 0.35±0.01 | 7.49±0.55(76) | 6.26±0.60 |
| Deep water (>1500db) | 8.18±0.25(142) | 7.79±0.04 | 0.37±0.02(142) | 1.03±1.29 | 8.98±0.39(142) | 7.60±0.21 |
| *DF4- Ligurian East* | | | | | | |
| surface water (0-150db) | 0.7±0.69(228) | 0.61±1.03 | 0.05±0.02(228) | 0.18±0.02 | 1.37±0.45(228) | 1.27±1.86 |
| LIW core ($S_{max}$ depth range: 300-500db) | 6.8±0.4(23) | 5.54±0 | 0.3±0.02(21) | 0.36±0.06 | 5.86±0.9(24) | 4.86±0 |
| Deep water (>1500db) | - | - | - | - | - | - |
| *DS2- Balearic Sea* | | | | | | |
| surface water (0-150db) | 1.32±1.46(196) | 1.19±1.5 | 0.08±0.04(196) | 0.11±0.04 | 1.61±0.64(196) | 1.54±0.78 |
| LIW core ($S_{max}$ depth range: 300-500db) | 8.32±0.32(58) | 6.92±0.12 | 0.37±0.02(60) | 0.39±0.003 | 7.31±0.9(60) | 7.55±0.62 |
| Deep water (>1500db) | 8.2±0.35(88) | - | 0.37±0.01(88) | - | 8.71±0.51(88) | 8.45±0.8 |
| *DF1- Algero-Provençal* | | | | | | |
| surface water (0-150db) | 0.87±0.85(372) | 1.08±1.7 | 0.05±0.02(372) | 0.07±0.05 | 1.42±0.3(372) | 1.28±0.73 |
| LIW core ($S_{max}$ depth range: 300-500db) | 8.07±0.34(126) | 7.51±0.18 | 0.36±0.02(126) | 0.34±0.008 | 6.84±0.95(126) | 5.96±0.77 |
| Deep water (>1500db) | 8.36±0.27(300) | 7.87±0.13 | 0.38±0.02(300) | 0.38±0.001 | 9.01±0.33(300) | 8.18±0.10 |
| *DS1- Alboran Sea* | | | | | | |
| surface water (0-150db) | 2.75±2.87(299) | 2.51±2.23 | 0.17±0.11(299) | 0.16±0.07 | 2.07±1.38(299) | 2.31±1.14 |
| LIW core ($S_{max}$ depth range: 400-600db) | 8.89±0.4(77) | 8.14±0.11 | 0.42±0.02(77) | 0.37±0.008 | 8.77±1.66(76) | 7.95±0.34 |
| Deep water (>1500db) | 7.72±0.81(65) | - | 0.36±0.04(65) | - | 8.98±0.63(65) | 8.16±0 |
| *DS3- Algerian West* | | | | | | |
| surface water (0-150db) | 1.8±1.88(254) | 1.82±2.01 | 0.11±0.05(354) | 0.11±0.06 | 1.71±0.68(354) | 2.10±0.91 |
| LIW core ($S_{max}$ depth range: 400-600db) | 9.33±0.08(70) | 8.28±0.15 | 0.41±0(73) | 0.38±0.012 | 8.1±0.53(72) | 6.68±0.80 |
| Deep water (>1500db) | 8.37±0.27(246) | 8.047±0.013 | 0.37±0.02(246) | 0.36±0.006 | 9.22±0.35(246) | 8.87±0.23 |
| *DS4- Algerian East* | | | | | | |
| surface water (0-150db) | 0.94±0.77(170) | 0.75±1.26 | 0.07±0.02(170) | 0.05±0.03 | 1.53±0.12(170) | 1.35±0.52 |
| LIW core ($S_{max}$ depth range: 400-600db) | 8.5±0.25(43) | 8.60±0.06 | 0.38±0.03(43) | 0.38±0.008 | 7.27±0.67(42) | 7.092±0.55 |
| Deep water (>1500db) | 7.94±0.24(132) | 8.06±0.06 | 0.36±0.02(132) | 0.38±0.006 | 8.73±0.38(132) | 9.04±0.24 |
| *DT1- Tyrrhenian North* | | | | | | |
| surface water (0-150db) | 1.03±1.14(231) | 0.88±1.2 | 0.06±0.02(231) | 0.09±0.03 | 1.64±0.52(231) | 2.19±0.59 |
| LIW core ($S_{max}$ depth range: 400-600db) | 5.95±0.49(43) | 5.86±0.36 | 0.27±0.03(44) | 0.308±0.02 | 7.06±0.08(44) | 6.76±0.59 |
| Deep water (>1500db) | 7.75±0.37(194) | 7.12±0.47 | 0.36±0.03(194) | 0.40±0.02 | 9.19±0.47(194) | 7.51±0.49 |
| *DT3- Tyrrhenian South* | | | | | | |
| surface water (0-150db) | 1.21±1.38(711) | 1.23±1.80 | 0.06±0.03(711) | 0.061±0.04 | 1.58±0.61(711) | 1.55±1.05 |
| LIW core ($S_{max}$ depth range: 300-500db) | 6.2±0.28(225) | 6.42±0.01 | 0.26±0.02(225) | 0.254±0.005 | 6.28±0.65(224) | 6.68±0.44 |
| Deep water (>1500db) | 7.88±0.4(227) | 7.12±0.26 | 0.37±0.02(227) | 0.31±0.007 | 9.04±0.52(227) | 8.02±0.07 |
| *DI1- Sardinia Channel* | | | | | | |
| surface water (0-150db) | 1.22±1.39(271) | 1.42±1.95 | 0.07±0.03(271) | 0.064±0.03 | 1.57±0.68(271) | 1.39±1.01 |
| LIW core ($S_{max}$ depth range: 300-500db) | 6.52±0.17(89) | 6.45±0.22 | 0.27±0.02(89) | 0.250±0.01 | 6.36±0.67(89) | 6.27±0.70 |
| Deep water (>1500db) | 7.91±0.62(107) | - | 0.37±0.03(107) | 0.32±0 | 8.64±0.91(107) | - |
| *DI3- Sicily Strait* | | | | | | |
| surface water (0-150db) | 0.87±0.68(583) | 0.77±0.81 | 0.06±0.02(583) | 0.063±0.02 | 1.53±0.29(583) | 1.44±0.58 |
| LIW core ($S_{max}$ depth range: 200-400db) | 4.95±0.47(80) | 5.14±0.14 | 0.21±0.02(78) | 0.194±0.004 | 5.26±0.79(81) | 6.744±0.41 |
| Deep water (>1500db) | - | - | - | - | - | - |

**Average (Avg) ± standard deviation of inorganic nutrient (the number observation within depth range) for three layers from the adjusted/new product and MEDATLAS vertical climatological profiles (called here Medar). Regions are defined according to Manca et al. (2004) (table 2S, Fig.2S)

