# Peer review of "Dissolved Inorganic Nutrients in the Western Mediterranean Sea (2004-2017)"

_Earth System Science Data, 2019_

## Short Comment (SC1) · 26 Oct 2019

To study nutrient dynamics in the western Mediterranean basin, it would have been more scientifically relevant to include all existing data, particularly those in the central Gulf of Lion, which undergo drastic changes each winter due to winter convection and the nutrients replishment into the surface (Testor et al., 2018; Kessouri et al., 2017). This is particularly unfortunate given that since 2010 the MOOSE observing network has been conducting the MOOSE_GE cruises (Coppola et al., 2019; Tintoré et al., 2019) every year, which samples more than 100 nutrient stations in the northwestern basin. These data are also available free of charge and in free access (SISMER; DOI 10.18142/235).

[Figure]

It is obvious that publishing your own data is quite logical but at the present time when we are talking about integrated and interoperable systems in Europe, this "forgetting" is regrettable, especially if we want to fully understand the dynamics of nutrients in the Western Mediterranean Sea.

Coppola, L., P. Raimbault, L. Mortier, and P. Testor (2019), Monitoring the environment in the northwestern Mediterranean Sea, Eos, 100, https://doi.org/10.1029/2019EO125951. Published on 25 July 2019.

Kessouri, F., Ulses, C., Estournel, C., Marsaleix, P., D'Ortenzio, F., Severin, T., et al. (2018). Vertical mixing effects on phytoplankton dynamics and organic carbon export in the western Mediterranean Sea. Journal of Geophysical Research: Oceans, 123, 1647–1669. https://doi.org/10.1002/ 2016JC012669

Testor, P., Bosse, A., Houpert, L., Margirier, F., Mortier, L., Legoff, H., . . . Conan, P. (2018). Multiscale observations of deep convection in the northwestern Mediterranean Sea during winter 2012–2013 using multiple platforms. Journal of Geophysical Research: Oceans, 123. https://doi.org/10.1002/2016JC012671

Tintoré J, Pinardi N. (2019) Challenges for Sustained Observing and Forecasting Systems in the Mediterranean Sea. Front. Mar. Sci. 6:568. doi: 10.3389/fmars.2019.00568

---

## Referee Comment (RC1) · Toste Tanhua (Referee) · 19 Nov 2019

This paper takes a systematic approach, using proven methods, to create an internally consistent data product of inorganic nutrient observations in the Western Mediterranean Sea. The work addresses a major issue with marine inorganic nutrient observations, biases in the data. This is a solid work and deserves to be published. However I have a few major issues with the work that I would like to see addressed, and a few minor comments.

The data product is available in a repository and can be easily downloaded. The same goes for the unadjusted data, all collected in a single file. This certainly has a lot of value. It seems to be a two-step procedure with data flagged as questionable (i.e. flag

"3") removed from the adjusted product. For cruises where the whole cruise was considered as of "poor quality" (as assessed from excessive scatter etc.) are still included in the product, but flagged as questionable. Why include questionable cruises in the product at all? I appreciate having access to the original data (i.e. prior to adjustment), but that does not preclude the need to link to the individual cruise files. These can be in a common format on a dedicated place, or it could be links to the original data file in a repository (NODC, SeaDataNet, or similar). That has value since for instance some of these cruises probably have associated "other" data, such as oxygen etc. that might be of use for the user. I recommend to establish links to the original data files.

This last comment does also go for the meta-data of the cruises. I guess in most cases this would include reference to a cruise report. I could not find any such references, please add links to cruise reports.

For the secondary QC, the authors choose to adjust all data to 5 reference cruises that was considered to have particularly high quality. One of the reasons was the well-known issue with bias in nutrient measurements being introduced by freezing of samples and analyzing them on land post-cruise. However, not all reference cruises had nutrient measured on-board. Why then include them as reference cruises?

Although it seems that the low-nutrient water of the Mediterranean might be less prone to bias due to freezing, the result from this study seem to suggest something different with all three variables being adjusted preferentially upward or downward. That might be an interesting result. Or maybe this is a function of bias in the measurements?

It would be useful to have a directory of crossover plots for all cruises. The method of GLODAP and CARINA could be taken as an example, but a repository on the web where the crossoverplots can be downloaded would go a long way. This would allow users to judge the validity of the adjustments.

Why only discuss a selection of cruises in section 5.4 ? All cruises had adjustments. I recommend to expand this section to cover all cruises.
Minor comments: Line 67: I suggest changing "profiles" for "observations"

Line 116: The CARIED data product is not yet published and available.

Line 115: Please refer to the GO-SHIP nutrient manual.

Line 188: I am not sure if this is a useful metric. The authors discuss the influence having observations in different sub-basins have on this statistic later. Why not create statistic that is for sub-basin by sub-basin?

Line 221: The $2°$ influence radius is probably fine for the Atlantic Ocean, but mostly not for the Mediterranean Sea. How did the author handle cross-overs that were influenced by observations from nearby other sub-basins where a different nutrient concentration could be expected?

Line 226: If you know that the deep water is (potentially) changing fast, why include it in the crossover analysis? Would it work to have a crossover analysis covering, for instance, $1000 - 2000$ meter only? If so, why was that not used, and how did the authors remove temporal natural variability of deep water?

Line 260: Here you decide not to include cruises that could not be adjusted in the product. On the other hand, you do include data that had only questionable data in the product (although flagged as such). Why? An alternative approach could be to include the data with a flag that indicate that the data did not undergo 2nd QC.

Section 6, and possibly elsewhere: GLODAP and CARINA are data products, rather than datasets. The difference being that the products have an additional layer of QC (2nd QC bias adjustment) applied, whereas a data set is a collection of data that are in its original form, possibly with consistent primary QC, unit conversion etc. Not so important perhaps, but a little of semantic difference.

Line 427: Not a complete list of authors for this paper.

Table 2: Why have a different format for this table compared to table 1?
Table 4: It would be useful to include the reference cruises in this table

Table 5: not a big deal, but the "*" sign in this table is applied in column 2, whereas in other tables (4) it is applied to column 1.

[Figure]

---

## Short Comment (SC2) · 19 Nov 2019

This data-product is a great addition to the existing Mediterranean data products, filling gaps in time and space in respect to nutrient data in a "key" region. The paper itself is further a great presentation of this new data-product. Used methods are extensively outlined such that the end-user can easily understand the assigned adjustments and flags. Furthermore, the methods follow best-known practices as already used and verified in other recommended data-products (e.g. Crossover-analysis used in a.o. GLODAP). It is important to note that the original data are published as well, thus

allowing later modifications/improvements of methods. Apart from one methodological issue concerning the calculated internal consistency no "big" content-related problems caught my attention. Finally, considering the understandable critics concerning the "incompleteness" of this data-product by not including new and existing MOOSE_GE cruises covering the central Golf of Lion, I would counteract that I see this product rather as a starting point of a living data product. I completely agree that in the next version the mentioned data should most definitely be included, but I would not deny the scientific value of this product. However, of course I have hopes for a living data-product being updated regularly as this is the only way to truly achieve a complete understanding of the nutrient dynamics in the med. Thanks to the authors, originators and everybody who has contributed!

Scientific Issues:

1) I have only detected one critical issue in this paper - the internal consistency calculation(s) and the resulting improvements. I am afraid that the weighted mean (WM) has been calculated wrongly (the used formula is correct!). One must use the "absolute" offsets in the WM-equation, i.e. an offset of 1.02 equals an "absolute" offset of 2%, as it does for an offset of 0.98. Thus in this example, the 2% in turn needs to be used in the WM-equation and not 1.02 or 0.98. Otherwise, these might cancel each other out. This also implies that the lower the WM the better, e.g. a WM of 2% is better than a WM of 3% in terms of internal consistency. This WM reflects the absolute weighted mean offset of the data set compared to the reference data set, hence the smaller the WM the higher the internal consistency. Consequently, lines 316 – 318 are misinterpretations of "wrong" numbers (crossover method application vs interpretation of the internal consistency)... The improvements should still be visible, however the calculations should be redone and the "correct" numbers should be given.

2) Concerning one minor issue - I am missing a presentation of the temporal distribution of the samples, i.e. the dates of the cruises. This would be nice to have to allow the detection of seasonal biases of the data-product (observational seasonal biases are

hard to surpass and totally fine themselves but should be mentioned if present).

3) Lines 228 – 230 might be misleading: Weights are given according to the "confidence" in the determined offset of the compared profiles not necessarily the variance of the profiles themselves. I.e. the weighted mean offset of a given crossover-pair is weighted to the depth where the offsets of all compared profiles have the smallest variation. Usually this is the case in deep regions and yes, this indeed is strongly interlinked with the degree of variance of each profile... Please consider rephrasing.

---

## Referee Comment (RC2) · Marina Lipizer (Referee) · 5 Jan 2020

General comments: This paper addresses the important issue of providing quality controlled datasets of biogeochemical parameters, based on 24 cruises carried out between 2004 – 2017 in the Western Mediterranean by the Italian National Research Council. The objective is relevant, since good quality biogeochemical data are fundamental to study temporal and spatial variability of oceanographic processes and the possible effects of global changes. In its current form, however, the paper presents several issues which need to be addressed in order to allow publication.

Concerning data availability, it is useful to get access to both the original dataset and the adjusted dataset, to allow users to reprocess the original data with different references, for example. The dataset is complete, with most required metadata, and provided in a user-friendly format. However, the adjusted dataset does not follow exactly WOCE QC flags: missing values are not flagged, while they should be flagged 9 (no data).

The dataset can provide a valuable contribution to the main European initiative in charge of assembling and giving access to marine data of the European seas, namely the European Marine Observation and Data network (EMODnet) (see Giorgetti et al., 2018). Surprisingly, there is no reference to the large availability of data in the Western Mediterranean provided by European data infrastructures such as SeaDataNet (https://www.seadatanet.org/) and EMODnet Chemistry (https://www.emodnet-chemistry.eu/).

The general approach for data Quality Control has already been used for a World Ocean dataset to achieve internal consistency of data and it is a solid method. However, I am concerned with the choice of the 5 cruises as reference to perform the secondary quality control and the adjustments, given the well known mesoscale dynamics of the Western Mediterranean, the seasonal variability detected also in the deep layers and the changes observed in the deep waters reported in the same period (Manca et al., 2004; Schroeder et al., 2008; Schroeder et al., 2016). It is recommended to compare the profiles of the reference cruises with the outcomes of the extensive analysis of over 40 years of biogeochemical data collected in the Mediterranean and the resulting climatological vertical profiles (Manca et al., 2004) and the full set of spatially averaged vertical profiles available to downloaded at http://nettuno.ogs.trieste.it/medar/climatologies/medz.html), provided for different Mediterranean regions defined according to general circulation patterns.

Giorgetti et al., 2018 EMODnet Chemistry Spatial Data Infrastructure for marine observations and related information. Ocean and Coastal Management 166 (2018) 9–17

Manca et al., 2004 Physical and biochemical averaged vertical profiles in the Mediterranean regions: an important tool to trace the climatology of water masses and to validate incoming data from operational oceanography; Journal of Marine Systems 48, 83–116

Schroeder et al. (2008) An extensive western Mediterranean deep water renewal between 2004 and 2006, Geophys. Res. Lett., 35, L18605, doi:10.1029/2008GL035146.

Specific comments:

To improve the logical sequence of the information, some sections should be reorganized.

The Introduction is not logically organized, there are several citations which are listed, but the connections are not clear. Many important concepts are introduced (eg. Biological pump, N:P ratio) but not introduced and some sentences are not clear or vague (eg. Lines 57- 60). Reference should also be made to the Mediterranean Sea – Eutrophication and Ocean Acidification aggregated datasets 1911/2017 v2018 provided by EMODnet Chemistry (https://doi.org/10.6092/89576629-66d0-4b76-8382-5ee6c7820c7f (line 71)

The use of citations should be revised: some citations do not seem to be appropriate or are not correctly inserted in the text as there are cases of quite vague statements linked to citations (eg. line 57 Boyd; Line 171: Muniz et al 2001)

Reference to published climatologies of biogeochemical properties available for the Mediterranean is missing (eg. Manca et al., 2004; MEDAR/MEDATLAS Climatology)

Section 2.2 should be moved after 4.1.

Section 3 should follow 2.1, after the description of sampling protocols for nutrient measurements.

Line 47: the latter: do you mean validation? Can you please explain what you mean?

Lines 83-88: there is a not correct comparison among different terms: datasets,

databases and large European data infrastructures such as SeaDataNet and EMOD-net Chemistry are different things.

Throughout the manuscript: check the consistency between the terms dataset and data set.

Is the description at lines 123-130 innovative? If not, the citation to the already consolidated method is enough and the whole part can be removed. On the other hand, a table summarising the laboratories, the instruments, the respective detection limits, together with sample storage and freezing duration used for the different cruises would facilitate the understanding.

Section 4: deals with Quality Assurance rather than QC

Section 4.1 should be reorganized to clearly explain how primary QC has been carried out; lines 169 – 172: please explain how were QF assigned to data and the relationships between flagging and CV

Lines 176-179: this sentence is not very clear. Please rephrase it.

Lines: 187-206: As shown in fig. 9, most cruises (even cruises #1, 5 and 16) cover different parts of the West Mediterranean basin, which are influenced by heterogeneous physical and biogeochemical processes, different water masses, which are characterised by different nutrient concentrations. The relationship between standard deviation of data collected in different water masses and data precision is not so straightforward. Therefore, the assessment of "precision of each cruise measurements" based on cruise CV is questionable.

The authors use 5 reference cruises carried out in different seasons between 2001 and 2016 to adjust data obtained during a total of 24 cruises carried out between 2004 and 2017. Reference cruises cover a large area but sometimes with just 1 station per sub-area. The use of single stations, sampled during a specific season as reference is questionable. Even though only data below 1000 m are involved in the Secondary QC

and deep waters are less variable than upper and intermediate waters, seasonal as well as long term variability in nutrient concentrations in deep waters cannot be ruled out, as also stated by the authors. It is not clear how this is taken into account (lines 226-230).

Section 5.4: why only a sub-set of cruises is described?

Line 373: Apart from old MEDAR/MEDATLAS database, reference should be made to the harmonized, aggregated and validated Mediterranean regional dataset of parameters related to eutrophication provided by EMODnet Chemistry (https://doi.org/10.6092/89576629-66d0-4b76-8382-5ee6c7820c7f)

Line 326-327: r2 do not match those in the figures

Line 577: N:P does not match those in the figures A plot showing temporal distribution of cruises and of reference cruises could be appreciated

Fig.1 Map: difficult to identify the different (Blue and red) cruises. The use of larger and filled/open symbols may help.

Fig.3: Numbers in figures do not match with captions.

Fig. 4: What are the codes "C1" and "C2"?

Fig. 8: Numbers in figures do not match with captions. Has the adjustment been done on the whole profile or only to data > 1000m? This is not clearly described in the paper.

References:

The first reference is not complete (journal? Pages?)

Line 427-429: check punctuation

Line 491-494: check punctuation

Finally, a careful language editing is required.

---

## Author Comment (AC1) · 31 Mar 2020

On behalf of all authors, we would like to thank the reviewer for their thorough reading of the manuscript and their constructive remarks and suggestions. Your comments provided valuable insights to refine and clarify the manuscript. We have taken into consideration all suggestions. In the following, we try to address all issues raised as best as possible.

R: It seems to be a two-step procedure with data flagged as questionable (i.e. flag "3") removed from the adjusted product. For cruises where the whole cruise was considered as of "poor quality" (as assessed from excessive scatter etc.) are still included in the product but flagged as questionable. Why include questionable cruises in the

product at all?

A: In the adjusted product, flags were based on the results of the 2nd QC, so "flag 3: Adjusted and recommended questionable", is a flag based on 2nd QC recommendation in section 4.4. i.e. It is a layer of flags in the final product for "flag 2: adjusted and acceptable" and "flag 3: adjusted and recommended questionable". We did clarifyy better it in Table 3 and in the supplementary Materials (Supplementary material – Part 2 (A2)). As mentioned in section 4.4, we have done the evaluation of the cruise overall quality but leave it up to the users how to appropriately use these data.

R: I appreciate having access to the original data (i.e. prior to adjustment), but that does not preclude the need to link to the individual cruise files. These can be in a common format on a dedicated place, or it could be links to the original data file in a repository (NODC, SeaDataNet, or similar). That has value since for instance some of these cruises probably have associated "other" data, such as oxygen etc. that might be of use for the user. I recommend to establish links to the original data files.This last comment does also go for the meta-data of the cruises. I guess in most cases this would include reference to a cruise report. I could not find any such references, please add links to cruise reports.

A: We agree that it is important to have easy access to the original, individual data files and metadata. Some of the cruise metadata such as cruise reports are available on http://www.seaforecast.cnr.it/reports/, but not all. We will add cruise reports for the missing cruises and submit all the individual cruise files to the SeaDataNet repository.

R: For the secondary QC, the authors choose to adjust all data to 5 reference cruises that was considered to have particularly high quality. One of the reasons was the well-known issue with bias in nutrient measurements being introduced by freezing of samples and analyzing them on land post-cruise.

A: We have modified the text to state that this is one of the criteria, but not a requirement for being a reference cruise.
R: However, not all reference cruises had nutrient measured on-board. Why then include them as reference cruises?

A: As reference cruises, we use only cruises that are known to have followed best practice standards, where nutrient analysis followed the recommendation of the World Ocean circulation experiment (WOCE) and the GO-SHIP protocols, and have undergone rigorous quality control following GLODAP routines or in the framework of the MedSHIP programme. We believe that observations of these cruises are of high degree of reliability, independently if the analysis was made on-board or on-land.

R: Although it seems that the low-nutrient water of the Mediterranean might be less prone to bias due to freezing, the result from this study seem to suggest something different with all three variables being adjusted preferentially upward or downward. That might be an interesting result. Or maybe this is a function of bias in the measurements??

A: We agree it could be due to bias in the measurement, we did not generalize it to all cruises. We tried to understand and find out what was the source of bias in the observations and the storage time was one of them. Freezing is not the main cause of the bias if samples were well preserved and unfrozen. One of the main reasons for the upward and downward biases would be the lack of use of Reference Material for Nutrients in those cruises as also noted in CARINA (Tanhua, T., Brown, P. J., and Key, R. M.: CARINA: nutrient data in the Atlantic Ocean, Earth Syst. Sci. Data, 1, 7–24, https://doi.org/10.5194/essd-1-7-2009, 2009. ) or the most recent global comparability exercise (Aoyama, M.: Global certified-reference-material- or reference-material-scaled nutrient gridded dataset GND13, Earth Syst. Sci. Data, 12, 487–499, https://doi.org/10.5194/essd-12-487-2020, 2020)

R: It would be useful to have a directory of crossover plots for all cruises. The method of GLODAP and CARINA could be taken as an example, but a repository on the web where the crossoverplots can be downloaded would go a long way. This would allow

users to judge the validity of the adjustments.

A: Yes, we want to make available the crossover plots following the crossover and adjustment Data Repository for CARINA or GLODAP, however it cannot be done easily, before the paper is published, we will work on making it available with the cruise reports.

R: Why only discuss a selection of cruises in section 5.4? All cruises had adjustments. I recommend expanding this section to cover all cruises.

A: We have expanded this section to cover all cruises.

R: Minor comments: Line 67: I suggest changing "profiles" for "observations

A: Done, in the revised version.

R: Line 116 The CARIMED data product is not yet published and available

A: We added a sentence about CARIMED data product (not yet available). The CARIMED initiative lead by M. Álvarez is a compilation of carbon and carbon relevant data for the MedSea that is taking longer than expected to be published, hopefully in 2020.

R: Line 115: Please refer to the GO-SHIP nutrient manual

A: Done.

R: Line 188: I am not sure if this is a useful metric. The authors discuss the influence having observations in different sub-basins have on this statistic later. Why not create statistic that is for sub-basin by sub-basin?

A: This point was raised by referee#2 as well. The standard deviation of data deeper than 1000db was defined as a first assessment to get indications about the precision of the measurements in each cruise following (Olsen et al., 2016). Statistics in different sub-basins has been added to check all cruises that have measurements in different

subregions (Table 4).

R: Line 221: The 2_ influence radius is probably fine for the Atlantic Ocean, but mostly not for the Mediterranean Sea. How did the author handle crossovers that were influenced by observations from nearby other sub-basins where a different nutrient concentration could be expected?

A: The reviewer is correct that we did not separate the analysis by sub-basin. The choice of the 2° was also partly for practical reasons since the number of reference cruises is too low to allow to restrict this radius. If we had more reference cruises, we could have reduced the 2° influence radius, but given that we only have 5, a relatively large influence radius is the only way to ensure statistically relevant results.

R: Line 226: If you know that the deep water is (potentially) changing fast, why include it in the crossover analysis? Would it work to have a crossover analysis covering, for instance, 1000 – 2000 meter only? If so, why was that not used, and how did the authors remove temporal natural variability of deep water?

A: The minimum chosen depth was 1000m, so that all cruises and all areas could be included in the 2nd QC and considering the relative low variability of the deep layer, compared to the intermediate and surface layers (nitrate CV=1.16, phosphate CV=1.005, silicate CV=0.75) the deep (>1000 db) layer (nitrate CV=0.15, phosphate CV=0.22, silicate CV=0.14)). The toolbox we use is not designed and tested to do crossover analyses for a part of the water column (e.g. 1000-2000m as suggested). It can only do it from Xm to the bottom. It would be possible to rewrite the code to do this, but that is beyond the scope of this paper, since we still aim to obtain results that could compare with other regions of the world ocean where the same method has been applied. The crossover analysis is done in density space. Thus, natural variability in the physics and water mass structure is accounted for in the method. Besides, we have a minimum adjustment limit for a reason and part of that reason is that we should not overcorrect when there is natural variability (which is always there).

R: Line 260: Here you decide not to include cruises that could not be adjusted in the product. On the other hand, you do include data that had only questionable data in the product (although flagged as such). Why? An alternative approach could be to include the data with a flag that indicate that the data did not undergo 2nd QC.

A: Yes, in the final product we included only cruises that underwent a 2nd QC, that is why we removed those that were not subjected to 2ndQC, those cruises are still in the original data collection. We did prefer to leave it up to the users how to appropriately use these data.

R: Section 6, and possibly elsewhere: GLODAP and CARINA are data products, rather than datasets. The difference being that the products have an additional layer of QC (2nd QC bias adjustment) applied, whereas a data set is a collection of data that are in its original form, possibly with consistent primary QC, unit conversion etc. Not so important perhaps, but a little of semantic difference.

A: We have changed this in the text, thank you for this important remark.

R: Line 427: Not a complete list of authors for this paper. A: The reference has been corrected.

R: Table 2: Why have a different format for this table compared to table 1? A: We have modified Table 2 to be comparable to Table 1.

R: Table 4: It would be useful to include the reference cruises in this table A: Table of the reference cruises is in the supplementary materials Table 1S, and we have added the number of samples.

R: Table 5: not a big deal, but the "*" sign in this table is applied in column 2, whereas in other tables (4) it is applied to column 1. A: The notation has been revised.

---

## Author Comment (AC2) · 31 Mar 2020

On behalf of all authors, we would like to thank the reviewer for their thorough reading of the manuscript and their constructive remarks and suggestions. Your comments provided valuable insights to refine and clarify the manuscript. We have taken into consideration all suggestions. In the following, we try to address all issues raised as best as possible.

R : The dataset is complete, with most required metadata, and provided in a user-friendly format. However, the adjusted dataset does not follow exactly WOCE QC flags: missing values are not flagged, while they should be flagged 9 (no data).

A : We did follow the WOCE QC flag during the 1st QC in the original dataset, "flag 9"

for missing or non-measured values. As for the adjusted product, we added flags based on the results of the crossover analysis excluding the non-measured one, so there is no "flag 9" in the adjusted product, there is only "flag 2 : adjusted and acceptable" and "flag 3 : adjusted and recommended questionable", this based on 2ndQC recommendation in section 4.4. We did clarify it better in table 3 and in the supplementary Materials (Supplementary material – Part 2 (A2)).

R : The dataset can provide a valuable contribution to the main European initiative in charge of assembling and giving access to marine data of the European seas, namely the European Marine Observation and Data network (EMODnet) (see Giorgetti et al., 2018). Surprisingly, there is no reference to the large availability of data in the Western Mediterranean provided by European data infrastructures such as SeaDataNet (https://www.seadatanet.org/) and EMODnet Chemistry (https://www.emodnet-chemistry.eu/).

A : We are aware about the large availability of data in the Western Mediterranean provided by European data infrastructures such as SeaDataNet (https://www.seadatanet.org/) and EMODnet Chemistry. We have now added a reference to the well-known existing nutrient datasets or data products. However, the main purpose of the paper is to make available the CNR data set. In our future studies, we aim at updating and adding other data sources from SeaDataNet and the MOOSE observing network, like Dr. Coppola suggested (Coppola et al., 2019; Tintoré et al., 2019), and integrating the Eastern Mediterranean as well, as far as possible.

R : I am concerned with the choice of the 5 cruises as reference to perform the secondary quality control and the adjustments, given the well known mesoscale dynamics of the Western Mediterranean, the seasonal variability detected also in the deep layers and the changes observed in the deep waters reported in the same period (Manca et al., 2004; Schroeder et al., 2008; Schroeder et al., 2016). It is recommended to compare the profiles of the reference cruises with the outcomes of the extensive analysis of over 40 years of biogeochemical data collected

in the Mediterranean and the resulting climatological vertical profiles (Manca et al., 2004) and the full set of spatially averaged vertical profiles available to downloaded at http://nettuno.ogs.trieste.it/medar/climatologies/medz.html), provided for different Mediterranean regions defined according to general circulation patterns. Giorgetti et al., 2018 EMODnet Chemistry Spatial Data Infrastructure for marine observations and related information. Ocean and Coastal Management 166 (2018) 9–17 Manca et al., 2004 Physical and biochemical averaged vertical profiles in the Mediterranean regions: an important tool to trace the climatology of water masses and to validate incoming data from operational oceanography; Journal of Marine Systems 48, 83–116 Schroeder et al. (2008) An extensive western Mediterranean deep-water renewal between 2004 and 2006, Geophys. Res. Lett., 35, L18605, doi:10.1029/2008GL035146.

A : Reference cruise data were chosen according to a number of criteria: they are independent from our CNR dataset, they have a large spatial distribution and different time span (we added information about number of observation per reference cruises table 2), Besides, nutrient analysis followed the recommendation of the World Ocean circulation experiment (WOCE) , the GO-SHIP protocols (Hydes et al., 2010; Tanhua et al., 2013) and have undergone rigorous quality control following GLODAP routines, along with cruises that were carried out in the framework of the MedSHIP programme (Schroeder et al., 2015). Observations of these cruises are highly reliable. Based on that, the 5 reference cruises were selected to perform the 2nd QC analysis, as explained in section 2.3. We have added additional details in the text explaining our choice of reference cruises. This work is a starting point for a living data product, the original data collection is available to improve the method/make updates. We have added a section comparing our results to the Manca et al. (2004) vertical climatological profiles in section 4.5, thank you for the suggestion, which we think is a great addition to the paper.

R : To improve the logical sequence of the information, some sections should be reorganized. The Introduction is not logically organized, there are several citations which

are listed, but the connections are not clear. Many important concepts are introduced (eg. Biological pump, N:P ratio) but not introduced and some sentences are not clear or vague (eg. Lines 57- 60).

A : The text has been revised for structure and flow. We thank the reviewer for suggesting the additional references which have now been included.

R : Reference should also be made to the Mediterranean Sea – Eutrophication and Ocean Acidification aggregated datasets 1911/2017 v2018 provided by EMODnet Chemistry (https://doi.org/10.6092/89576629-66d0-4b76-8382- 5ee6c7820c7f) (line 71)

A : Done

R : The use of citations should be revised: some citations do not seem to be appropriate or are not correctly inserted in the text as there are cases of quite vague statements linked to citations (eg. line 57 Boyd; Line 171: Muniz et al 2001)

A : The text has been proofread to ensure proper citation throughout.

R : Reference to published climatologies of biogeochemical properties available for the Mediterranean is missing (eg. Manca et al., 2004; MEDAR/MEDATLAS Climatology)

A : Done, by comparing the product to Medar/MEDATLAS Climatology

R : Section 2.2 should be moved after 4.1. Section 3 should follow 2.1, after the description of sampling protocols for nutrient measurements.

A : The sections have been modified accordingly.

R : Line 47: the latter: do you mean validation? Can you please explain what you mean?

A: We meant data collection and monitoring, and there are still gaps in the Mediterranean Sea, this is now explained better in the text and refer to dataset and data product previously done.

R : Lines 83-88: there is a not correct comparison among different terms: datasets, databases and large European data infrastructures such as SeaDataNet and EMODnet Chemistry are different things. Throughout the manuscript: check the consistency between the terms dataset and data set.

A : we have made the necessary changes.

R : Is the description at lines 123-130 innovative? If not, the citation to the already consolidated method is enough and the whole part can be removed.

A : It is a summary of the analysis description done in laboratory.

R : On the other hand, a table summarising the laboratories, the instruments, the respective detection limits, together with sample storage and freezing duration used for the different cruises would facilitate the understanding.

A : A table of the sample storage and freezing is added in the revised supplementary materials (Table 1S).

R : Section 4: deals with Quality Assurance rather than QC

A : The section title has been modified accordingly

R : Section 4.1 should be reorganized to clearly explain how primary QC has been carried out; lines 169 – 172: please explain how were QF assigned to data and the relationships between flagging and CV Lines 176-179: this sentence is not very clear. Please rephrase it.

A: Section 4.1 has been reorganized as requested. The CV compares the degree of variation between surface and deep observations and how we can proceed with the flagging. The upper layer ( nitrate CV=1.16, phosphate CV=1.005, and silicate CV=0.75) imposed a check of outliers per depth range, here we name it as standard depths (or class of depth) at 0-10, 10-30, 30-60, 60-80, 80-160, 160-260, 260-360, 360-

460, 460-560, 560-1000 m. Per cruise, a Median absolute deviation was computed by class of depth, atypical observation was flagged as questionable, in the upper layer we did not strictly follow the criteria of flagging as bad the values higher than three median absolute deviations. The deep observation is relatively less variable (nitrate CV=0.15, phosphate CV=0.22, silicate CV=0.14).and the flagging was based on a check of nitrate to phosphate (N:P) and nitrate to silicate (N: Si) ratios. We considered as outlier any value that departs from the median ratio (below 1000db) by more than three median absolute deviations. We did highlight that the primary QC can be subjective depending on the expertise of the person flagging the data, thus flagging could bring in some uncertainties.

R : Lines: 187-206: As shown in fig. 9, most cruises (even cruises #1, 5 and 16) cover different parts of the West Mediterranean basin, which are influenced by heterogeneous physical and biogeochemical processes, different water masses, which are characterised by different nutrient concentrations. The relationship between standard deviation of data collected in different water masses and data precision is not so straightforward. Therefore, the assessment of "precision of each cruise measurements" based on cruise CV is questionable.

A : In order to have a first assessment of the precision of each cruise measurements, the standard deviation of data deeper than 1000db was calculated (Table 4). When the time span between different cruises is one month or less, then the temporal variation can be excluded below 1000db, and the standard deviation is interpreted as the effect of the natural variability and the precision of the observations. We compared standard deviations of cruises having similar spatial coverage. We add statistics per subregion as an overview of the overall content in nutrient layer (Table 4).

R : The authors use 5 reference cruises carried out in different seasons between 2001 and 2016 to adjust data obtained during a total of 24 cruises carried out between 2004 and 2017. Reference cruises cover a large area but sometimes with just 1 station per sub-area. The use of single stations sampled during a specific season as reference is

questionable. Even though only data below 1000 m are involved in the Secondary QC and deep waters are less variable than upper and intermediate waters, seasonal as well as long term variability in nutrient concentrations in deep waters cannot be ruled out, as also stated by the authors. It is not clear how this is taken into account (lines 226-230).

A: A valid crossover with the reference cruises should consider at least three stations for the computation to get a valid statistic. If there are not enough stations, there is no crossover. The computational approach takes this into account if there is a change, weights are given according to the "confidence" in the determined offset of the compared profiles not necessarily the variance of the profiles themselves, i.e. the weighted mean offset of a given crossover-pair is weighted to the depth where the offsets of all compared profiles have the smallest variation, which is the case in deep regions. The summary plot coming out of the 2nd QC toolbox is a function of year. The reason for that is to be able to assess whether a long-term trend exists (it becomes very obvious when the offsets are plotted as a function of time). So long-term trends are taken into account when deciding on an adjustment. Seasonal variability is an issue though, but this is why we go deep.

R : Section 5.4: why only a sub-set of cruises is described?

A : The section has been expanded to discuss all cruises.

R : Line 373: Apart from old MEDAR/MEDATLAS database, reference should be made to the harmonized, aggregated and validated Mediterranean regional dataset of parameters related to eutrophication provided by EMODnet Chemistry (https://doi.org/10.6092/89576629-66d0-4b76-8382-5ee6c7820c7f)

A : We did add it in the revised version.

R : Line 326-327: r2 do not match those in the figures, Line 577: N:P does not match those in the figures

A : We checked it and corrected it in the revised version.

R : A plot showing temporal distribution of cruises and of reference cruises could be appreciated

A : Table 1 is the cruise summary table where cruises were sorted in chronological order and plotted against the reference in Figure 1. Same as table 2 detailing the reference cruises and plotted in figure 2.

R : Fig.1 Map: difficult to identify the different (Blue and red) cruises. The use of larger and filled/open symbols may help.

A : We improved the map.

R : Fig. 8: Numbers in figures do not match with captions. Has the adjustment been done on the whole Profile or only to data > 1000m? This is not clearly described in the paper.

A : The proposed adjustment factor was estimated from observation deeper than 1000db, and we applied it to the whole profile. We clarified it in the revised version (section 3.2).

R : References: The first reference is not complete (journal? Pages?)

A : Done.

---

## Author Comment (AC3) · 31 Mar 2020

On behalf of all authors, we would like to express our thanks for your thorough reading of the manuscript and suggestions and for the positive comments, as you mentioned, the new product, is a starting point for us for a living data collection.

1) lines 316 – 318 are misinterpretations of "wrong" numbers (crossover method application vs interpretation of the internal consistency) . . . The improvements should still be visible; however, the calculations should be redone, and the "correct" numbers should be given.

Thank you for pointing the issue with calculation, it was corrected in the final version of the paper.

[Figure]

2) Concerning one minor issue - I am missing a presentation of the temporal distribution of the samples, i.e. the dates of the cruises. This would be nice to have to allow the detection of seasonal biases of the data-product (observational seasonal biases are hard to surpass and totally fine themselves but should be mentioned if present).

In all figures and tables, cruises are in chronological order, that's why we did not add samples distribution.

3) Lines 228 – 230 might be misleading: Weights are given according to the "confidence" in the determined offset of the compared profiles

Done.

---

## Referee Report (RR1)

**General comments:**

The presentation of the results and the paper in general has improved since the first submission. All requested comments have been taken into consideration.

**Specific comments:**

There are still some language weaknesses throughout the text. Eg. line 20-22: "an extensive dataset " which require English editing, before the paper can be published.

There are only format checks to be done: the use of commas and spaces must be corrected in some citations, in citing figures and tables, in the reference list:

eg. Line 70; 72; 137; 677; 731-733

Once format corrections are completed, the paper can be published.

---

## Referee Report (RR2)

**Review of the resubmitted manuscript "Dissolved Inorganic Nutrients in the Western Mediterranean Sea (2004-2017)" by Belgacem et al.**

I thank the authors for responding to my requests. Below are some comments on the replies.

===============

*I appreciate having access to the original data (i.e. prior to adjustment), but that does not preclude the need to link to the individual cruise files. These can be in a common format on a dedicated place, or it could be links to the original data file in a repository (NODC, SeaDataNet, or similar). That has value since for instance some of these cruises probably have associated "other" data, such as oxygen etc. that might be of use for the user. I recommend to establish links to the original data files.*
*This last comment does also go for the meta-data of the cruises. I guess in most cases this would include reference to a cruise report. I could not find any such references, please add links to cruise reports.*
We agree that it is important to have easy access to the original, individual data files and metadata. Some of the cruise metadata such as cruise reports are available on http://www.seaforecast.cnr.it/reports/, but not all. We will add cruise reports for the missing cruises and submit all the individual cruise files to the SeaDataNet repository.
===========

The response is good, and I accept that having the individual cruise files at SeaDataNet is an acceptable solution. However, this is only stated as an intention, nowhere in the manuscript do I see any mention to the seaforecast site nor to SeaDataNet. Before the article can be published these links and the content within those links needs to be established.
* * *
*Although it seems that the low-nutrient water of the Mediterranean might be less prone to bias due to freezing, the result from this study seem to suggest something different with all three variables being adjusted preferentially upward or downward. That might be an interesting result. Or maybe this is a function of bias in the measurements??*
We agree it could be due to bias in the measurement, we did not generalize it to all cruises. We tried to understand and find out what was the source of bias in the observations and the storage time was one of them. Freezing is not the main cause of the bias if samples were well preserved and unfrozen. One of the main reasons for the upward and downward biases would be the lack of use of Reference Material for Nutrients in those cruises as also noted in CARINA (Tanhua, T., Brown, P. J., and Key, R. M.: CARINA: nutrient data in the Atlantic Ocean, Earth Syst. Sci. Data, 1, 7–24, https://doi.org/10.5194/essd-1-7-2009, 2009. ) or the most recent global comparability exercise (Aoyama, M.: Global certified-reference-material- or reference-material-scaled nutrient gridded dataset GND13, Earth Syst. Sci. Data, 12, 487–499, https://doi.org/10.5194/essd-12-487-2020, 2020)
* * *
I do not see any evidence in the manuscript that support your statement that "Freezing is not the main cause of the bias if samples were well preserved and unfrozen". Instead I see statements related to silicate that freezing does have an impact. I know that there is (anecdotal) indications that the freezing of samples does not affect the low nutrient waters of the Mediterranean as much as other basins. I was asking for a short discussion of this in the manuscript. Why are all adjustments in Figure 5 downward? Same for Figure 7.
* * *
*It would be useful to have a directory of crossover plots for all cruises. The method of GLODAP and CARINA could be taken as an example, but a repository on the web where the crossoverplots can be downloaded would go a long way. This would allow users to judge the validity of the adjustments.*

Yes, we want to make available the crossover plots following the crossover and adjustment Data Repository for CARINA or GLODAP, however it cannot be done easily, before the paper is published, we will work on making it available with the cruise reports.

Same thing here, I do not think that stating an intention is sufficient.

*Line 221: The 2_ influence radius is probably fine for the Atlantic Ocean, but mostly not for the Mediterranean Sea. How did the author handle crossovers that were influenced by observations from nearby other sub-basins where a different nutrient concentration could be expected?*

The reviewer is correct that we did not separate the analysis by sub-basin. The choice of the 2° was also partly for practical reasons since the number of reference cruises is too low to allow to restrict this radius. If we had more reference cruises, we could have reduced the 2° influence radius, but given that we only have 5, a relatively large influence radius is the only way to ensure statistically relevant results.

This is not a satisfactory answer. I can accept the 2° radius of influence, but not that you are potentially comparing observations in two different sub-basins.

---

## Author Response (AR2)

Dear Editor, we thank you for your (and referees') time spent helping us to improve the manuscript. We particularly appreciate the effort in this period of pandemic closures when running even standard business has become complicated for all.

Please find uploaded the new version, with all remarks by the Editor and Referee 2 implemented. In particular, we reopened a ticket on PANGAEA asking for the modifications requested by the Editor.

(i)     Column headers in CNR_DIN_WMED_20042017_original.csv and CNR_DIN_WMED_20042017_adjusted.csv  are now the very same (ii)    We uploaded a new readme file with more information and in particular with  links to the reports and the cruise PIs along with explanation of the abbreviations. Abbreviations are explained accordingly in the supplementary material section A2 "Data product organization details".

Please be aware that as of today (16/06/2020), the ticket to PANGAEA is not yet resolved and we are waiting for PANGAEA feedbacks for final implementation.

We also implemented all minor edits requested. Please note that we added in the manuscript a new table, listing data sources and links to the reports (accessed June 2020). This new table has been numbered Table 1b. Accordingly, the old table 1 is now table 1a.

[revised manuscript text omitted]

**Supplementary material – Part 2**

**A1.**  **Data product** description

The  data product includes 870 stations sampled during 24 cruises between 2004 and 2017 in the Western Mediterranean Sea  mainly on board of research vessels owned by the Italian National Research Council. It includes bottle data combined with CTD data.

In all stations, measurements were carried out with a CTD-rosette system consisting of a CTD SBE 911 plus and a General Oceanics rosette with 24 12-l Niskin Bottles at the observed depth of the bottle sample. Temperature measurements were performed with an SBE-3/F thermometer with a resolution of $10{-}3$ °C and conductivity measurements were performed with an SBE-4 sensor with a resolution of $3 \cdot 10{-}4$ S/m. The probes were calibrated before and after the cruise. Except for salinity, no certified reference material (CRM) was used. CTD salinity was calibrated against measurements made with a salinometer.

Samples of nitrate, phosphate and silicate were frozen to -20°C and stored before being analysed in laboratories onshore.

Measurements were subjected to a rigorous quality control (primary and secondary quality control) and the dataset presented is the product adjusted after the application of quality control approaches.

**A2.**  **Data product** organization details

Cruise identification: To guarantee the comparability between measurements, an alphanumeric identification code (ID) together with an expedition code (Expocode) are defined a unique identifier. The list of the parameters included in the data product are detailed in table below:

| # |  Short name in data files | Parameter | Unit/format | Method/ description | Comment | Original Dataset | Adjusted |
|---|---|---|---|---|---|---|---|
| 1 | EXPOCODE | Expedition code |  24 EXPOCODEs | 12 digits ; *Shipcode_yyyy_mm_dd* *yyyy_mm_dd: cruise starting day* | | ✓ | ✓ |
| 2 | CRUISE | Cruise ID | 24 Cruise IDs |  | | ✓ | ✓ |
| 3 | DATE | Event date | yyyy-mm-dd | | | ✓ | ✓ |
| 4 | TIME | Event time | hhmm | | | ✓ | ✓ |
| 5 | DAY | Day | dd | | | ✓ | ✓ |
| 6 | MONTH | Month | mm | | | ✓ | ✓ |
| 7 | YEAR | Year | yyyy | | | ✓ | ✓ |
| 8 | LATITUDE | Longitude | | | | ✓ | ✓ |
| 9 | LONGITUDE | Latitude | | | | ✓ | ✓ |
| 10 | STNNBR | Station number | | | | ✓ | ✓ |
| 11 | BTLNBR | Niskin bottle number | | | | ✓ | ✓ |
| 12 | CASTNO | Cast number | | | | ✓ | ✓ |
| 13 | CTDPRS | Pressure | dbar | CTD pressure | | ✓ | ✓ |
| 14 | DEPTH | Depth | Meters | Depth from pressure | | ✓ | ✓ |
| 15 | CTDSAL | Salinity | | CTD salinity | PSS-78 | ✓ | ✓ |
| 16 | CTDSAL_FLAG_W | Salinity flag | | WOCE flags | | ✓ | ✓ |
| 17 | CTDTMP | Temperature | °C | CTD temperature | ITS-90 | ✓ | ✓ |
| 18 | THETA | Potential temperature | | Theta from CTDTMP & CTDSAL | | ✓ | ✓ |
| 19 | NITRAT | Nitrate | $\mu$mol kg$^{-1}$ | standard colorimetric methods* | | ✓ | ✓ |
| 20 | NITRAT_FLAG_W  | Nitrate flag  | | WOCE flags After 1st quality control / Flags after 2nd QC Flag 2: adjusted and acceptable Flag 3: adjusted and recommended questionable | Details in Section 4.4 | ✓ | ✓ |
| 21 | PHSPHT | Phosphate | $\mu$mol kg$^{-1}$ | standard colorimetric methods* | | ✓ | ✓ |
| 22 | PHSPHT_FLAG_W  | Phosphate flag  | | WOCE flags After 1st quality control / Flags after 2nd QC Flag 2: adjusted and acceptable Flag 3: adjusted and recommended questionable | Details in Section 4.4 | ✓ | ✓ |
| 23 | SILCAT | Silicate | $\mu$mol kg$^{-1}$ | standard colorimetric methods* | | ✓ | ✓ |
| 24 | SILCAT_FLAG_W  | Silicate flag  | | WOCE flags After 1st quality control / Flags after 2nd QC Flag 2: adjusted and acceptable Flag 3: adjusted and recommended questionable | Details in Section 4.4 | ✓ | ✓ |

\* Standard colorimetric methods of seawater analysis (Grasshoff et al. (1999)).

- **Data format**

*Original dataset: CNR_DIN_WMED_20042017_original.csv:* This is the original dataset with 24  parameter including flag variables of 24 cruises for nitrate, phosphate, silicate and CTD salinity from the primary quality control.

*Adjusted dataset: CNR_DIN_WMED_20042017_adjusted.csv*: This is the adjusted product with 24 parameter, after removing outlier data (issued from primary quality control) and after applying adjustment factors from the secondary quality control (Crossover Analysis).

---

## Author Response (AR3)

Dear Editor,

Thank you very much for the good news of acceptance. As requested, dataset modifications are now implemented in PANGAEA and the official DOI is assigned. PANGAEA DOI is then updated in the current version of the manuscript.

Best regards

Jacopo Chiggiato

CNR-ISMAR

[revised manuscript text omitted]